# PACT: Self-Evolving Physical Safety Alignment for Diffusion Policies in Embodied Manipulation

**Lingxuan Wu** [1 2]  **Zijian Zhu** [1]  **Lizhong Wang** [1]  **Chengyang Ying** [1]  **Huayu Chen** [1]  **Xiao Yang** [1]  **Fangming Liu** [2]
**Jun Zhu** [1]

## Abstract

Diffusion policies have achieved remarkable success in robotic manipulation, yet they often fail to satisfy strict physical constraints required for safe deployment. Existing approaches impose safety either prematurely during training or reactively via external guardrails at test time, limiting policy expressivity and overall scalability. We propose Physical safety Alignment for Constrained Trajectories (PACT), a self-evolving post-training framework that projects pretrained diffusion policies onto constraint-feasible regions without accessing demonstration data or task rewards. PACT distills constraint gradients into the diffusion model through a reverse-KL objective with dense supervision across timesteps. It incorporates a curriculum that progressively tightens constraints while maintaining theoretically bounded policy shift and monotone improvement, mitigating the safety-performance trade-off from catastrophic forgetting. On simulated and real-world embodied manipulation benchmarks, PACT significantly reduces safety violations by 31.0% on average while improving task success by 30.7%.[1]

## 1. Introduction

Recent advances in diffusion policies have demonstrated remarkable progress in robotic manipulation (Janner et al., 2022; Pearce et al., 2023; Chi et al., 2025), enabling expressive, multimodal action distributions and strong generalization across diverse tasks and environments (Liu et al., 2025;

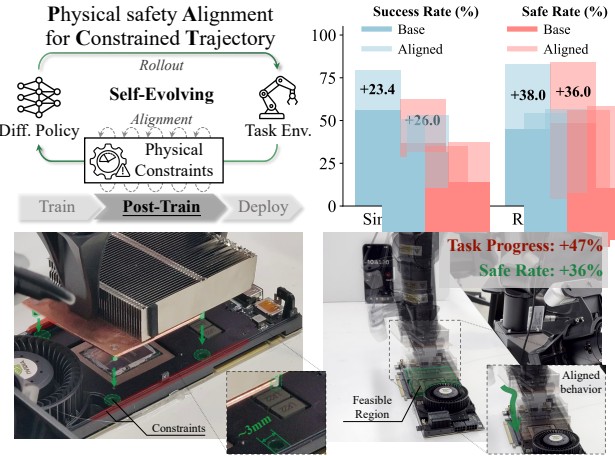

*Figure 1.* **Physical safety alignment for diffusion-based manipulation.** PACT aligns a pretrained diffusion policy in a post-training stage using self-rollouts and continuous constraint supervision throughout the diffusion process in a *self-evolving manner without external demonstrations or rewards*. *Top right:* PACT improves both task performance and safety in simulation and real-world settings, resolving the safety–performance trade-off. *Bottom:* GPU assembly requiring millimeter-level precision, PACT enables emergent fine-grained correction behaviors that adjust trajectories into feasible regions, ensuring accurate and safe insertions.

Amin et al., 2025). Despite their power, real-world deployment requires strict adherence to physical constraints, such as collision avoidance, force limits, and kinematic feasibility (Rueß, 2022). Constraint violations can be catastrophic, leading to task failure, irreversible hardware damage, and human injury, especially in safety-critical applications like manufacturing (Buhl et al., 2019) and surgery (Hu et al., 2023). Consequently, deploying diffusion policies at scale necessitates preserving their expressive power while guaranteeing constraint satisfaction under all operating conditions.

To mitigate this issue, existing methods can be broadly categorized into design-time (or train-time) and test-time approaches. Design-time approaches (Ross et al., 2011) embed safety constraints during data collection (Liu et al., 2024; Bahety et al., 2025) or model training (García & Fernández, 2015; Ciftci et al., 2025), ensuring robustness to perturbations during execution. However, they require extensive human expertise and task-specific engineering,

---

[1]Dept. of Comp. Sci. and Tech., Institute for AI, Tsinghua-Bosch Joint ML Center, THBI Lab, BNRist Center, Tsinghua University, Beijing, 100084, China [2]Peng Cheng Laboratory, 518108, China. Correspondence to: Xiao Yang <yangxiao19@tsinghua.org.cn>, Jun Zhu <dcszj@tsinghua.edu.cn>.

*Proceedings of the 43rd International Conference on Machine Learning*, Seoul, South Korea. PMLR 306, 2026. Copyright 2026 by the author(s).

[1]Code: https://github.com/thu-ml/PACT.

limiting scalable deployment. Test-time approaches (Hsu et al., 2023) instead employ guardrails (Reichlin et al., 2022; Deng et al., 2025) or backup policies to intercept unsafe actions during execution (Wong et al., 2022). However, they depend on predefined constraints or additional sensors for real-time monitoring. Therefore, current methods impose safety constraints either prematurely, restricting training and potentially impeding learning, or reactively through limited deployment-stage interventions. Both strategies compromise policy expressivity and scalability while failing to provide principled safety guarantees for deployment.

Drawing inspiration from post-training safety alignment in large language models (Bai et al., 2022; Dai et al., 2024; Chen et al., 2024), we align pretrained diffusion policies for physical safety before deployment. We formulate it as projecting pretrained policies onto the feasible physically constrained regions in a Constrained Markov Decision Process (CMDP) (Altman, 2021). This preserves the expressiveness of the original policies, enables flexible adaptation to varying constraint sets without retraining, and maintains computational efficiency by decoupling safety from capability learning. However, safety alignment for embodied manipulation (Billard & Kragic, 2019) presents unique challenges. First, diffusion models inherently spread probability mass across the action space, complicating strict constraint satisfaction even after alignment. Second, post-training cannot access demonstration data or task utility functions, precluding conventional fine-tuning through data recollection (Liu et al., 2024) or reward-based optimization (García & Fernández, 2015). Third, aggressive safety enforcement risks inducing catastrophic forgetting (McCloskey & Cohen, 1989), since restricting policy support may eliminate task-critical behaviors when safe and original distributions diverge. Collectively, these challenges require scalable post-training mechanisms that enforce physical constraints while preserving the expressivity and the task performance.

To address these challenges, we propose **Physical safety Alignment for Constrained Trajectories (PACT)**, a self-evolving post-training framework that aligns diffusion policies with physical safety constraints. First, PACT introduces constraint distillation to tackle mode concentration by densely injecting gradient signals from safety cost functions at every diffusion timestep. This is equivalent to minimizing the backward KL divergence that concentrates the policy distribution into feasible regions while mitigating exposure bias (Bengio et al., 2015). In contrast to RL methods that suffer from sparse or noisy rewards, PACT provides stable, continuous supervision throughout the entire diffusion process. Second, self-evolving optimization overcomes the data accessibility challenge by operating exclusively on self-generated trajectories, eliminating dependencies on external demonstrations or reward engineering. Third, we incorporate a curriculum-based distillation that progressively tight-

ens safety constraints to mitigate the performance–safety trade-off (Zhang et al., 2025). This curriculum theoretically promotes bounded policy shift and monotonic improvement, preventing abrupt action collapse and preserving task competence by avoiding catastrophic forgetting. Together, these components constitute a scalable post-training alignment framework that embeds physical safety into diffusion policies through curriculum-based constraint distillation on self-collected rollouts, substantially reducing human effort by eliminating the need for demonstrations, task-specific utility functions, or outcome annotations.

We evaluate the policy aligned using PACT across simulated and real-world manipulation benchmarks that require delicate bimanual coordination, *without privileged state information or explicit constraint computation* at test time. PACT reduces safety violation rates by 31.0% while improving task success rates by 30.7% and achieving higher training efficiency compared to RL-based safety alignment methods. Notably, PACT exhibits strong real-world performance, including challenging GPU assembly tasks requiring millimeter-level precision, validating PACT as a practical paradigm for deploying diffusion policies in safety-critical robotic systems. Our contributions are as follows:

- To our knowledge, we introduce the first post-training framework for aligning diffusion-based manipulation policies with physical safety constraints, requiring no manual data collection or test-time guardrails.

- We develop a constraint distillation framework with curriculum-based updates that preserves task performance during safety enforcement in the absence of task utilities or demonstrations, enabling stable on-policy optimization with provable, continuous safety improvement.

- Our method substantially reduces safety violations while improving task success rates and training efficiency relative to RL-based alignment, and demonstrates real-world performance on challenging manipulation tasks.

## 2. Background

**Diffusion and flow policies.** Explicitly modeling stochasticity and multi-modality, diffusion-based policies (Sohl-Dickstein et al., 2015; Ho et al., 2020) enable more expressive control representations and have proven effective for real-world robotic tasks involving diverse behaviors (Janner et al., 2022; Chen et al., 2023; Chi et al., 2025). Given an action $\boldsymbol{a}$, a forward corruption process gradually injects Gaussian noise with a specific noise schedule $\alpha_t$ and $\sigma_t$:

$$\boldsymbol{a}_t = \alpha_t \boldsymbol{a} + \sigma_t \boldsymbol{\epsilon}, \quad \boldsymbol{\epsilon} \sim \mathcal{N}(0, \boldsymbol{I}), \ t \in [0, 1]. \quad (1)$$

Intuitively, diffusion policies are learned by training a denoising model $\boldsymbol{\epsilon}_\phi$ that predicts the injected noise $\boldsymbol{\epsilon}$:

$$\min_\phi \ \mathbb{E}_{t,\epsilon,\boldsymbol{s},\boldsymbol{a}\sim\mu(\cdot|\boldsymbol{s})}\big[\|\boldsymbol{\epsilon}_\phi(\boldsymbol{a}_t, \boldsymbol{s}, t) - \boldsymbol{\epsilon}\|_2^2\big], \quad (2)$$

which implicitly recovers the score function of the diffused behavior distribution (Song et al., 2021c):

$$\nabla_{\boldsymbol{a}_t}\mu_t(\boldsymbol{a}_t \mid \boldsymbol{s}, t) \approx \nabla_{\boldsymbol{a}_t}\mu_\phi(\boldsymbol{a}_t \mid \boldsymbol{s}, t) = \boldsymbol{\epsilon}_\phi(\boldsymbol{a}_t, \boldsymbol{s}, t)/\sigma_t, \tag{3}$$

and allows sampling actions from the learned behavior policy $\mu_\phi$. Beyond discrete denoising formulations, flow policies parameterize the policy as a velocity field that defines a deterministic flow transporting samples from noise to the action distribution (Liu et al., 2023), proven to be an equivalent continuous-time interpretation through probability flow ODEs (Lipman et al., 2023; Song et al., 2021a). Below we view diffusion and flow-based behavior policies as a unified generative policy class (Kingma & Gao, 2023) for subsequent physical safety alignment.

**Physical safety enforcement.** Prior work on physical safety in robotic manipulation (Haddadin, 2015) can be categorized by the stage at which safety is enforced. Development-time approaches incorporate safety during data collection or policy learning (Brunke et al., 2022), primarily through imitation learning (IL) (Ross et al., 2011; Yang et al., 2024; Liu et al., 2024; Bahety et al., 2025) and safe reinforcement learning (Safe RL) (García & Fernández, 2015; Altman, 2021; Ying et al., 2022). IL-based methods imitate curated safe demonstrations but suffer from degraded performance due to restricted action support. Safe RL formulates safety as auxiliary cost functions or constrained objectives, and typically relies on task utility to drive learning (Altman, 2021; Ying et al., 2022; Lee et al., 2022). However, the rewards are sparse and delayed in real-world settings (Gu et al., 2024), resulting in poor performance. Inference-time methods enforce safety through action filtering or control barrier functions that intervene upon constraint violations (Reichlin et al., 2022; Wabersich et al., 2023; Deng et al., 2025; Jung et al., 2025; Nakamura et al., 2025). However, these approaches require privileged state access, additional sensors, or perception models (Wong et al., 2022; Gokmen et al., 2023), thereby increasing inference-time overhead and limiting applicability to novel scenarios. Post-training methods, recently explored by SafeVLA (Zhang et al., 2026), amortize safety supervision into the policy at the intermediate stage between capability acquisition and deployment. However, it is limited to discrete categorical distribution and low-dimensional navigation tasks. We address *diffusion policies* operating over *continuous, high-dimensional* action spaces with complex physical safety constraints for *manipulation*, posing substantial challenges.

# 3. Methodology

## 3.1. Problem Formulation

We study *physical safety alignment* of diffusion policies under explicit safety constraints. The environment is modeled as a *constrained Markov Decision Process* (CMDP)

$\mathcal{M} = \langle \mathcal{S}, \mathcal{A}, \mathcal{P}, \mathcal{C} \rangle$ (Altman, 2021), where $\mathcal{S}$ and $\mathcal{A}$ respectively denote the state and action spaces, $\mathcal{P}(\boldsymbol{s}_{t+1} \mid \boldsymbol{s}_t, \boldsymbol{a}_t)$ denotes the transition dynamics, $\mathcal{C} = \{c_k(\boldsymbol{s}, \boldsymbol{a})\}_{k=1}^m$ is a set of safety cost functions encoding physical constraints (*e.g.*, collisions or excessive force). The *safe region* (feasible policy set) derived from safety costs $\mathcal{C}$ is defined as

$$\Pi_{\text{safe}} = \left\{ \pi : \mathbb{E}_{(\boldsymbol{s}, \boldsymbol{a}) \sim d^\pi} c_k(\boldsymbol{s}, \boldsymbol{a}) \leq d_k, \ \forall k \right\}. \tag{4}$$

Unlike standard CMDP formulations that rely on rewards, we frame *safety alignment* as a *regularized projection* of the base policy $\mu_\phi(\boldsymbol{a}_t \mid \boldsymbol{s}_t)$ onto the feasible set $\Pi_{\text{safe}}$:

$$\arg \min_{\pi \in \Pi_{\text{safe}}} \mathbb{E}_{\boldsymbol{s} \sim d^\pi} \left[ D_{\text{KL}}\big(\pi(\cdot \mid \boldsymbol{s}) \,\|\, \mu_\phi(\cdot \mid \boldsymbol{s})\big) \right], \tag{5}$$

which seeks the closest safety-compliant policy while preserving the behavioral fidelity of the original policy. Furthermore, we assume *no access to the pretrained behavior dataset* during safety alignment, which is realistic but restrictive (Dulac-Arnold et al., 2021). This precludes replay-based regularization or supervised correction to control distribution shift, making direct constrained optimization infeasible (Ball et al., 2023). To solve Problem (5), we leverage the Lagrange multiplier method to incorporate safety constraints while minimizing the KL divergence from the base policy (Chow et al., 2018; Tessler et al., 2019). Specifically, we introduce a set of Lagrange multipliers $\lambda_k \geq 0$ for each safety constraint $c_k(\boldsymbol{s}, \boldsymbol{a})$ which enforce the safety conditions in the optimization process. The Lagrangian $\mathcal{L}(\pi, \lambda)$ for the constrained optimization problem is formulated as:

$$\mathbb{E}_{(\boldsymbol{s}, \boldsymbol{a}) \sim d^\pi}\left[ D_{\text{KL}}\left( \pi(\cdot \mid \boldsymbol{s}) \| \mu_\phi(\cdot \mid \boldsymbol{s}) \right) + \sum_{k=1}^m \lambda_k (c_k(\boldsymbol{s}, \boldsymbol{a}) - d_k) \right], \tag{6}$$

which shares a similar structure with objective in Regularized RL (Dayan & Hinton, 1997; Schulman et al., 2017).

## 3.2. Physical Safety Alignment for Diffusion Policies

To minimize the objective in Eq. (6), extensive prior work has studied parametric policies such as Gaussian or categorical distributions using policy gradient (Schulman et al., 2017). However, extending these approaches to diffusion policies is non-trivial. In particular, policy gradient methods require tractable likelihood ratios or entropy terms, whereas diffusion policies only admit implicit densities whose likelihoods can only be computed approximately via costly probability ODE solvers or variational bounds for SDEs (Song et al., 2021b). These limitations make direct optimization of Eq. (6) impractical for diffusion models. Despite the intractability of likelihood-based optimization, the optimal solution to Eq. (6) admits a simple closed-form structure. For fixed Lagrange multipliers $\{\lambda_k\}_{k=1}^m$, the constrained optimal policy satisfies (Peng et al., 2019):

$$\pi^*(\boldsymbol{a} \mid \boldsymbol{s}) \ \propto \ \mu_\phi(\boldsymbol{a} \mid \boldsymbol{s}) \exp\left( -\sum_{k=1}^m \lambda_k \, c_k(\boldsymbol{s}, \boldsymbol{a}) \right), \tag{7}$$

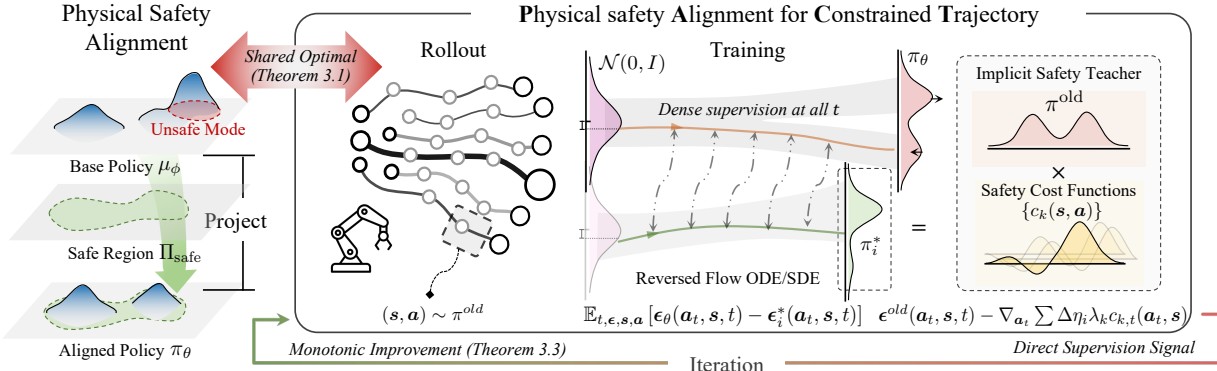

*Figure 2.* **Overview of Physical safety Alignment for Constrained Trajectories.** PACT frames physical safety alignment as projecting a pretrained diffusion policy $\mu_\phi$ onto the CMDP feasible set $\Pi_{\text{safe}}$ via a KL-regularized constrained objective. For fixed multipliers, the score function of the optimal aligned policy is defined as an *implicit safety teacher* by combining the base score with differentiable cost gradients (Theorem 3.1). We distill this teacher into a student diffusion policy $\pi_\theta$ using self-rollouts with direct and dense supervision across diffusion time and in a self-evolving manner without manual demonstrations or rewards. A curriculum schedule progressively increases constraint strength, yielding controlled policy shifts and monotonic safety improvement over iterations (Theorem 3.3).

which can be interpreted as an exponential tilting of the base policy by safety costs. Although the normalized density in Eq. (7) remains intractable, it is available for sampling as its score function is directly computable:

$$\boldsymbol{\epsilon}^*(\boldsymbol{a}_t, \boldsymbol{s}, t) \triangleq \boldsymbol{\epsilon}_\phi(\boldsymbol{a}_t, \boldsymbol{s}, t) - \sum_{k=1}^m \lambda_k \nabla_{\boldsymbol{a}_t} c_{k,t}(\boldsymbol{s}, \boldsymbol{a}_t)/\sigma_t, \quad (8)$$

where $c_{k,t}(\cdot)$ denotes the intermediate costs derived from the original cost function $c_k(\cdot)$ (Lu et al., 2023) with its detailed structure elaborated in Appx. A.1. We define the score function $\boldsymbol{\epsilon}^*(\cdot)$ in Eq. (8) as the *implicit safety teacher*.

However, sampling with $\boldsymbol{\epsilon}^*(\cdot)$ requires evaluating the cost functions $c_k$, which often depend on privileged environment states (e.g., object poses) typically obtained via additional sensors or auxiliary perception modules. This reliance hinders direct test-time deployment and is inherently unscalable, as the per-agent sensing and computation overhead grows with the size of the robot fleet. To overcome this limitation, we approximate $\boldsymbol{\epsilon}^*$ via distillation by training a student diffusion policy $\boldsymbol{\epsilon}_\theta$ to match the implicit teacher's score function for physical safety alignment[2]:

$$\min_\theta \mathbb{E}_{t,\boldsymbol{\epsilon},(\boldsymbol{s},\boldsymbol{a})\sim d^{\pi_\theta}} \|\boldsymbol{\epsilon}_\theta(\boldsymbol{a}_t, \boldsymbol{s}, t) - \boldsymbol{\epsilon}^*(\boldsymbol{a}_t, \boldsymbol{s}, t)\|^2. \quad (9)$$

**Theorem 3.1** (Optimality of PACT). *Given unlimited model capacity and sufficient data, the optimal solution for the distillation objective in Eq. (9) is the score function of Eq. (7)* [3].

*Remark* 3.2. This theorem shows that distillation can recover the optimal solution of the constrained optimization problem without explicit likelihood evaluation. Compared to RL-based methods that rely on stochastic, high-variance

---

[2]The objective in Eq. (9) is parameterization-agnostic. Further discussion is provided in Appx. A.2.

[3]Proof in Appx. A.3

reward signals, it provides direct and stable guidance toward the feasible region without requiring outcome rewards or value models. Moreover, the proposed objective is *solver-agnostic*, enabling efficient sampling with few denoising steps and higher-order solvers during rollout, unlike prior diffusion RL methods that are tightly coupled to first-order SDE samplers (Black et al., 2024; Liu et al., 2026a).

**Curriculum distillation to mitigate irreversible Out-of-Distribution collapse.** In practice, a key failure mode occurs when transient intermediate policies deviate substantially from the base behavior, inducing rapid rollout state distribution drift and triggering *Irreversible Out-of-Distribution (OOD) Collapse*, wherein an under-optimized policy is driven into OOD regimes and suffers a severe loss of task competence. This issue is exacerbated in safety alignment because task utility is inaccessible; therefore, no corrective supervision is available to steer recovery, making the degradation effectively irreversible. Mechanistically, it is primarily due to the lack of trust region control over intermediate updates (Conn et al., 2000) for direct distillation, so small deviations can compound over time and rapidly push trajectories into OOD states. To address this, we introduce a curriculum distillation strategy that progressively enforces safety constraints, enabling a smooth transition toward the final aligned solution. Concretely, we adopt a monotonically increasing curriculum schedule $\eta_0, \ldots, \eta_N \in [0, 1]$ with $\eta_0 = 0$ and $\eta_N = 1$, which gradually scales the constraint multipliers over $N$ iterations. At each iteration $i$, we solve the following intermediate objective:

$$\min_\pi \mathbb{E}_{(\boldsymbol{s},\boldsymbol{a})\sim d^\pi} \Big[ D_{\text{KL}}\big(\pi(\cdot \mid \boldsymbol{s}) \| \pi^{\text{old}}(\cdot \mid \boldsymbol{s})\big)$$
$$+ \sum_{k=1}^m \Delta\eta_i \cdot \lambda_k(c_k(\boldsymbol{s}, \boldsymbol{a}) - d_k) \Big], \quad (10)$$

where $\Delta\eta_i = \eta_i - \eta_{i-1}(i > 0)$ denotes the incremental

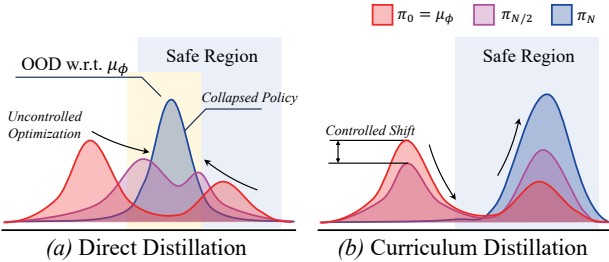

*(a) Direct Distillation*      *(b) Curriculum Distillation*

*Figure 3.* **Curriculum distillation mitigates Irreversible OOD Collapse by controlling intermediate policy shift.** We illustrate the evolution of policy distributions over iterations: *(a) Direct distillation* enforces constraints without control, so intermediate policies can drift rapidly, pushing rollouts into OOD regions and yielding a collapsed policy that loses task competence despite aiming for safety. *(b) Curriculum distillation* progressively increases constraint strength and regularizes policy within a trust region, inducing controlled transition to a safety-compliant policy.

schedule step. $\pi^{\text{old}}$ is the trained $\pi$ of the previous iteration, and $\pi^{\text{old}} = \mu_\phi$ at the initial iteration ($i = 1$). The corresponding distillation objective at iteration $i$ is:

$$\min_\theta \; \mathbb{E}_{t,\boldsymbol{\epsilon},(\boldsymbol{s},\boldsymbol{a})\sim d^{\pi_\theta}} \|\boldsymbol{\epsilon}_\theta(\boldsymbol{a}_t,\boldsymbol{s},t) - \boldsymbol{\epsilon}_i^+(\boldsymbol{a}_t,\boldsymbol{s},t)\|^2, \text{ with}$$

$$\boldsymbol{\epsilon}_i^+(\boldsymbol{a}_t,\boldsymbol{s},t) = \boldsymbol{\epsilon}^{\text{old}}(\boldsymbol{a}_t,\boldsymbol{s},t) - \sum_{k=1}^m \Delta\eta_i \, \lambda_k \nabla_{\boldsymbol{a}_t} c_{k,t}(\boldsymbol{s},\boldsymbol{a}_t)/\sigma_t,$$

$$(11)$$

where $\boldsymbol{\epsilon}_i^+(\cdot)$ defines the *Curriculum Implicit Safety Teacher*, serving as an intermediate target that smoothly interpolates toward the final Implicit Safety Teacher $\boldsymbol{\epsilon}^*(\cdot)$ in Eq. (8).

**Theorem 3.3** (Monotonic Improvement under Curriculum Distillation). *Assume each cost function is bounded, i.e., $c_k(\boldsymbol{s},\boldsymbol{a}) \in [c_k^{\min}, c_k^{\max}]$ for all $k$ and all $(\boldsymbol{s},\boldsymbol{a})$. Define the aggregated Lagrangian cost*

$$\ell(\boldsymbol{s},\boldsymbol{a}) \triangleq \sum_{k=1}^m \lambda_k \big(c_k(\boldsymbol{s},\boldsymbol{a}) - d_k\big), \quad \ell(\boldsymbol{s},\boldsymbol{a}) \in [\ell_{\min}, \ell_{\max}].$$

*Let $\pi^{new}$ be obtained by solving the objective in Eq. (11) at any iteration $i$; the update satisfies:*

- **Monotonic improvement.** *The expected curriculum Lagrangian cost under $d^{\pi_i}$ is non-increasing:*

$$\mathbb{E}_{d^{\pi^{new}}}\big[\ell(\boldsymbol{s},\boldsymbol{a})\big] \; \leq \; \mathbb{E}_{d^{\pi^{old}}}\big[\ell(\boldsymbol{s},\boldsymbol{a})\big]. \quad (12)$$

- **Controlled policy shift.** *Set $\Delta\ell = \ell_{\max} - \ell_{\min}$, then the policy change is bounded as*

$$\mathbb{E}_{\boldsymbol{s}\sim d^{\pi^{old}}}\Big[D_{\text{KL}}\big(\pi^{new}(\cdot \mid \boldsymbol{s})\|\pi^{old}(\cdot \mid \boldsymbol{s})\big)\Big] \leq \Delta\eta_i \Delta\ell. \quad (13)$$

Thus, the proposed curriculum distillation ensures monotonic safety improvement along the iterations and constructs a smooth policy with controlled policy shifts to prevent abrupt distributional shifts in the original objective.

---

**Algorithm 1** The Training Pipeline of PACT

**Require:** Pretrained diffusion policy $\boldsymbol{\epsilon}_\phi$, number of iterations $N$, number of epochs $E$, set of differentiable safety cost functions $\{c_k(\boldsymbol{s},\boldsymbol{a})\}_{k=1}^m$ and corresponding fixed Lagrange multipliers $\{\lambda_k\}_{k=1}^m$, curriculum schedule $\{\eta_i\}_{i=0}^N$, max diffusion time to inject cost guidance $t_c$.
**Ensure:** The aligned diffusion policy $\boldsymbol{\epsilon}_\theta$.
1: **for** iteration $i \leftarrow 1$ **to** $N$ **do**
2:      ▷ *Rollout and collect data*      ◁
3:      Rollout policy $\boldsymbol{\epsilon}_\theta$ and collect trajectories as $\mathcal{D} = \{(\boldsymbol{s},\boldsymbol{a})\}$.
4:      **for** epoch $\leftarrow 1$ **to** $E$ **do**
5:          **for all** mini-batch $\{\boldsymbol{s},\boldsymbol{a}\} \in \mathcal{D}$ **do**
6:              $\boldsymbol{a}_t = \alpha_t \boldsymbol{a} + \sigma_t \boldsymbol{\epsilon}$ with $\boldsymbol{\epsilon} \sim \mathcal{N}(0,\boldsymbol{I})$, $t \in [0,1]$.
7:              $\boldsymbol{a}_{0|t} = \big(\boldsymbol{a}_t - \sigma_t \boldsymbol{\epsilon}^{\text{old}}(\boldsymbol{s},\boldsymbol{a}_t,t)\big)/\alpha_t.$    (Eq.(1))
8:              Update $\theta$ by minimizing the objective in Eq. (11) with approximated score for Curriculum Implicit Safety Teacher computed with Eq. (15).
9:          **end for**
10:      **end for**
11:      Update $\theta_{\text{old}} \leftarrow \theta$ and clear buffer $\mathcal{D}_\tau \leftarrow \varnothing$
12: **end for**

---

**Corollary 3.4** (Curriculum PACT Optimal Solution). *The optimal solution $\pi_i^*$ at iteration $i$ satisfies*

$$\pi_i^*(\boldsymbol{a} \mid \boldsymbol{s}) \; \propto \; \mu(\boldsymbol{a} \mid \boldsymbol{s}) \exp\Big(-\sum_{k=1}^m \eta_i \cdot \lambda_k \, c_k(\boldsymbol{s},\boldsymbol{a})\Big), \; (14)$$

*and the final solution at iteration $N$ exactly coincides with the shared closed-form optimal solution in Theorem 3.1 given unlimited model capacity and data samples [4].*

In conclusion, the curriculum distillation procedure preserves the optimal solution of the original constrained objective. Its scheduled formulation ensures that each intermediate update enforces only a controlled deviation from the current policy with guaranteed improvement. This gradual enforcement of constraints prevents the policy from being prematurely driven into unfamiliar states, thereby avoiding Irreversible OOD Collapse and ensuring training stability.

### 3.3. Practical Implementations

**Training-free approximation for intermediate cost gradient and few-steps distillation.** The exact gradient of intermediate costs requires an additional denoising-time condition and training steps (Lu et al., 2023). However, injecting cost gradient only during the final few denoising steps of guided sampling is effective in most scenarios (Bao et al., 2022; Xu et al., 2023). Motivated by the few-step nature of guided sampling, we propose a training-free approximation to the intermediate cost gradient based on a Taylor expansion for $\nabla_{\boldsymbol{a}_t} c_{k,t}(\boldsymbol{s},\boldsymbol{a}_t)$ around $t = 0$. Specifically, for sufficiently small $t$, we approximate the gradient at $\boldsymbol{a}_t$ by evaluating it at the posterior mean of the clean action, defined as $\boldsymbol{a}_{0|t} \triangleq \mathbb{E}_{q(\boldsymbol{a}_0|\boldsymbol{a}_t,\boldsymbol{s})}[\boldsymbol{a}_0]$, yielding

---

[4]Proof in Appx. A.5

*Table 1.* **Task success and safety across bimanual manipulation benchmarks in RoboTwin.** We report *Success Rate (succ.)* and *Safe Rate (safe)* for each base policy before and after post-training with PACT.

| Model | Pick Dual Bottle | | Pick Diverse Bottle | | Handover Apple | | Handover Block | | Place Dual Shoes | | Pour Water | | Stack Blocks | | Average | |
|---|---|---|---|---|---|---|---|---|---|---|---|---|---|---|---|---|
| | Succ. | Safe | Succ. | Safe | Succ. | Safe | Succ. | Safe | Succ. | Safe | Succ. | Safe | Succ. | Safe | Succ. | Safe |
| DP | 67% | 36% | 36% | 6% | 63% | 64% | 28% | 70% | 23% | 66% | 38% | 20% | 63% | 24% | 45% | 41% |
| + PACT | **96%** | **89%** | **58%** | **21%** | **81%** | **82%** | **72%** | **78%** | **41%** | **73%** | **86%** | **64%** | **80%** | **48%** | **73%** | **65%** |
| DP3 | 44% | 24% | 4% | 2% | 54% | 46% | 59% | 71% | 9% | 59% | 16% | 13% | 6% | 6% | 27% | 32% |
| + PACT | **60%** | **46%** | **11%** | **7%** | **70%** | **59%** | **72%** | **73%** | **19%** | **70%** | **41%** | **16%** | **27%** | **16%** | **43%** | **41%** |
| RDT-1B | 61% | 27% | 35% | 12% | 76% | 52% | 64% | 61% | 42% | 72% | 52% | 17% | 60% | 13% | 56% | 36% |
| + PACT | **88%** | **74%** | **71%** | **43%** | **83%** | **59%** | **75%** | **82%** | **63%** | **82%** | **85%** | **72%** | **89%** | **24%** | **79%** | **62%** |
| $\pi_{0.5}$ | 64% | 20% | 37% | 8% | 71% | 55% | 66% | 64% | 41% | 70% | 48% | 14% | 67% | 25% | 56% | 37% |
| + PACT | **89%** | **72%** | **76%** | **42%** | **86%** | **61%** | **74%** | **79%** | **70%** | **79%** | **79%** | **72%** | **92%** | **30%** | **81%** | **62%** |

$\nabla_{a_t} c_{k,t}(s, a_t) \approx \nabla_{a_t} c_k (s, a_{0|t})$, where $q(a_0 \mid a_t, s)$ denotes the posterior for the clean action $a_0$ and $a_{0|t}$ can be simply computed by re-formatting Eq. (1) to a one-step reverse diffusion update (Chung et al., 2023). This approximation is widely used in training-free guided sampling across diverse tasks (Chung et al., 2023; Yu et al., 2023), including bimanual manipulation (Deng et al., 2025). The training-free approximation yields the few-step Curriculum Implicit Safety Teacher in Eq. (11) as:

$$
\epsilon_i^+(a_t, s, t) \approx
\begin{cases}
\epsilon^{\text{old}}(a_t, s, t) - \sum_{k=1}^m \Delta\eta_i \, \lambda_k \nabla_{a_t} c_k(s, a_{0|t}), & t < t_c, \\
\epsilon^{\text{old}}(a_t, s, t), & t \geq t_c.
\end{cases}
\tag{15}
$$

This requires only the original differentiable cost functions. By restricting constraint injection to the final few steps (e.g., $t_c = 0.03$ in all experiments), the approach substantially reduces the computational overhead from the Jacobian evaluations. The theoretical and empirical justifications are provided in Appx. A.6 and ablation studies in Sec. 4.

## 4. Experiments

### 4.1. Experimental Setup

**Physical constraints.** We adopt the unsafe-behavior taxonomy for bimanual manipulation from Deng et al. (2025), which encompasses the majority of hazard patterns identified through extensive case studies. To evaluate PACT, we select three representative constraint functions (i.e., Poking, Alignment, and Rotation) along with corresponding tasks (e.g., pouring water). The constraints require privileged environment states, which can be obtained directly through simulation APIs. For real-world experiments, we estimate these states using off-the-shelf 3D perception pipelines (Huang et al., 2025). See Appx. B for more details.

### 4.2. Simulation Evaluation

**Environments and tasks.** We evaluate these methods on bimanual manipulation tasks from RoboTwin (Mu et al.,

2024; Chen et al., 2025) with a *randomized* setup, requiring bimanual coordination and robust handling of safety-critical interactions. See Appx. C.1 & C.2 for details.

**Base policies and implementations.** To demonstrate generality, we apply PACT to multiple pretrained diffusion-based policies with a wide spectrum of model architectures, base capacities, input modalities, diffusion parameterizations, and sampling strategies: Diffusion Policy (DP) (Chi et al., 2025), 3D Diffusion Policy (DP3) (Ze et al., 2024), which takes point clouds as model input, and generalist policies like RDT-1B (Liu et al., 2025) and $\pi_{0.5}$ (Black et al., 2025a). During post-training, each policy collects data by rolling out on 1,000 training scenes. We perform full-parameter fine-tuning for DP and DP3, and apply Low-Rank Adaptation (LoRA) (Hu et al., 2022) to align RDT-1B and $\pi_{0.5}$. Additional details are provided in Appx. C.5.

**Baselines.** We organize baselines along two axes: learning paradigm (IL vs. RL) and data regime (off-policy vs. on-policy). Off-policy IL baselines include behavior cloning on intervened rollouts as in Probe, Learn, Distill (PLD) (Xiao et al., 2026) and cloning on rollouts collected with the implicit safe teacher (guided rollouts), while off-policy RL is represented by iDQL (Hansen-Estruch et al., 2023). To enable comparison between distillation and other objectives, we also evaluate offline distillation (Meng et al., 2023) with different types of rollouts. On-policy baselines use self-collected data and include IL methods (Rejection Fine-tuning (RFT) and online variant of PLD (PLD$_{\text{online}}$)) and RL methods including AWR (Peng et al., 2019), QSM (Psenka et al., 2024), DIPO (Yang et al., 2023), and PPO-style diffusion optimization (Schulman et al., 2017; Liu et al., 2026a). All methods are initialized from the same pretrained DP and are matched in environment interaction and update budgets. Details are provided in Appx. C.6 and Appx. C.7.

**Metrics.** We report *Success Rate*, measuring task completion, and *Safe Rate*, measuring the fraction of rollouts that satisfy *all* physical safety demands. The details on the metrics computation are elaborated in Appx. C.4.

*Table 2.* **Performance comparison with on-policy baselines.** Both the Success Rate (Succ.) and Safe Rate (Safe) are reported.

| Method | Pick Dual Bottles | | Handover Apple | | Pour Water | | Stack Blocks | |
|---|---|---|---|---|---|---|---|---|
| | Succ. | Safe | Succ. | Safe | Succ. | Safe | Succ. | Safe |
| *Imitation Learning* | | | | | | | | |
| Base | 67% | 36% | 63% | 64% | 38% | 20% | 63% | 24% |
| PLD$_{online}$ | 93% | 66% | 70% | 70% | 55% | 29% | 71% | 32% |
| *Reinforcement Learning* | | | | | | | | |
| PPO | 71% | 37% | 67% | 67% | 60% | 31% | 73% | 33% |
| DIPO | 77% | 43% | 65% | 68% | 67% | 29% | 66% | 36% |
| *Distillation* | | | | | | | | |
| Ours | **96%** | **89%** | **81%** | **82%** | **86%** | **64%** | **80%** | **48%** |

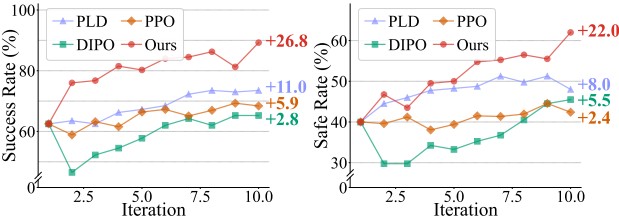

*Figure 4.* **Training efficiency comparison with on-policy baselines.** Success Rate (left) and Safe Rate (right), are averaged over four tasks across training iterations. Our method demonstrates the most training efficiency and stability.

**Effectiveness of PACT.** The aggregate performance across tasks is summarized in Table 1. Overall, post-training with PACT consistently improves both task completion and manipulation safety across all base policies. For instance, DP shows substantial gains in both success rate by 28% and safety rate by 24%. It indicates that the aligned behaviors become simultaneously more effective and more physically compliant, which facilitates the reconciliation between performance and safety (Zhang et al., 2026) through curriculum-based constraint distillation, which injects dense constraint gradients into the diffusion policy while progressively tightening safety enforcement to preserve behavioral fidelity. Furthermore, improvements are also more pronounced on difficult, precision-critical tasks (e.g., Pour Water and Stack Blocks), where minor misplacement can readily trigger constraint violations that cause failure. Qualitative results for simulated tasks are provided in Appx. E.2.

**Comparison to on-policy baselines.** We further compare PACT with representative on-policy baselines in Table 2. Across all four tasks, PACT attains the highest Success Rate and Safe Rate, indicating improved constraint satisfaction without sacrificing task performance. Beyond effectiveness, PACT is markedly more stable: several on-policy alternatives (e.g., AWR, RFT, and QSM) inevitably collapse to near-zero success in our setting (See Appx. E.5 for details), and RL-based baselines often require additional stabilization mechanisms (e.g., soft target updates in DIPO or KL regularization in PPO). In contrast, PACT leverages direct supervision from gradients of differentiable safety costs, avoiding the high-variance credit assignment inherent to

*Table 3.* **Performance comparison with off-policy baselines.** Success Rate (Succ.) and Safe Rate (Safe) are reported for each task. "RO" denotes abbreviation for Rollout.

| Method | Pick Dual Bottles | | Handover Apple | | Pour Water | | Stack Blocks | |
|---|---|---|---|---|---|---|---|---|
| | Succ. | Safe | Succ. | Safe | Succ. | Safe | Succ. | Safe |
| *Imitation Learning* | | | | | | | | |
| Base | 67% | 36% | 63% | 64% | 38% | 20% | 63% | 24% |
| Guided RO | 82% | 74% | 74% | 79% | 70% | 41% | 66% | **59%** |
| PLD | 70% | 43% | 70% | 72% | 54% | 47% | 71% | 40% |
| *Reinforcement Learning* | | | | | | | | |
| iDQL | 74% | 40% | 63% | 70% | 62% | 28% | 65% | 31% |
| *Distillation* | | | | | | | | |
| Expert RO | 70% | 41% | 71% | 68% | 67% | 24% | 65% | 27% |
| Self RO | 76% | 44% | 70% | 67% | 71% | 28% | 72% | 34% |
| Guided RO | 70% | 45% | 72% | 78% | 72% | 31% | 77% | 28% |
| Ours | **96%** | **89%** | **81%** | **82%** | **86%** | **64%** | **80%** | 48% |

policy-gradient updates whose quality is dominated by the sampled batch (Nota & Thomas, 2020). This stability is further reflected in Fig. 4, where PACT improves smoothly without the early drops observed in DIPO or the oscillations of PPO. Finally, PACT is the most efficient, converging more than 5× faster because constraint distillation provides dense, per-timestep supervision throughout the diffusion process. RL-based methods instead rely on coarse trajectory-level signals and costly, noisy likelihood estimation, and therefore require substantially more interaction to reach comparable gains. Additional results are provided in Appx. E.

**Comparison to off-policy baselines.** We compare PACT against off-policy alternatives spanning imitation-based approaches (Guided RO and PLD) and an RL-based method (iDQL). As shown in Table 3, PACT attains higher success and safety rates in most settings, with particularly large gains on *Pick Dual Bottles* and *Pour Water*. We also implement an off-policy distillation variant under different data sources. Distillation is especially effective on more complex tasks (e.g., *Pour Water* and *Stack Blocks*), where collecting sufficient rollouts for imitation is difficult. However, distribution tilting via distillation can leverage limited data more effectively. Moreover, off-policy variants of PACT underperform the on-policy iterative update (ours) under matched gradient budgets, underscoring the importance of on-policy with the reverse-KL objective in Eq. (5), which mitigates forgetting in utility-free circumstances (Shenfeld et al., 2026) and avoids the constraint leakage induced by the mode-covering behavior of forward-KL.

**Ablation Studies. 1) Curriculum distillation.** We first ablate the curriculum used to progressively tighten safety constraints during training. Specifically, we compare direct distillation (Eq. (9)) to curriculum distillation with a linear schedule $\eta_i = i/N$. We find that direct distillation induces catastrophic forgetting with significant performance drops (Fig. 13) in contrast with the smoother improvement dynamics yielded by curriculum-based enforcement (Fig. 4). **2)**

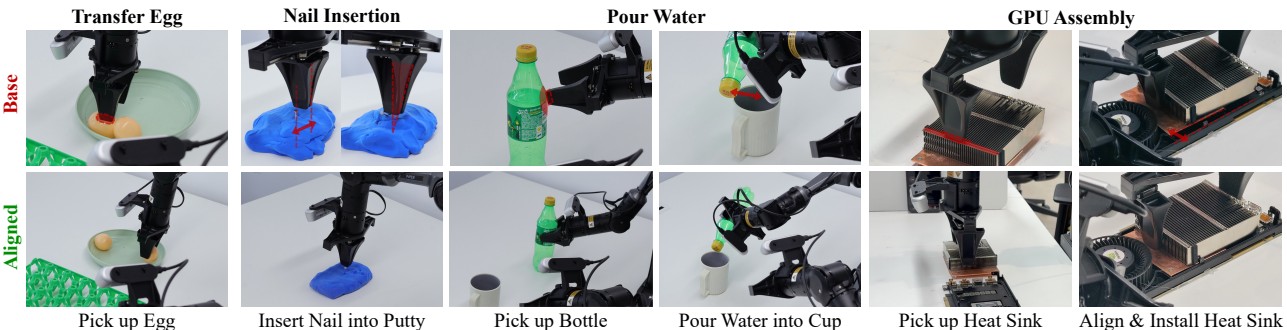

**Transfer Egg  Nail Insertion  Pour Water  GPU Assembly**

*Figure 5.* **Qualitative results of real world evaluation.** base policy (top) v.s. policy aligned by PACT (bottom) across four manipulation tasks. PACT reduces unsafe contacts and improves task completion by correcting key failure modes: avoiding poking to securely grasp the egg (*Transfer Egg*); aligning the gripper with the nail head, preventing lateral or tilted insertion (*Nail Insertion*); eliminating bottle poking and cup-rim misalignment (*Pour Water*); avoiding poking heat-sink and corrects installation misalignment (*GPU Assembly*).

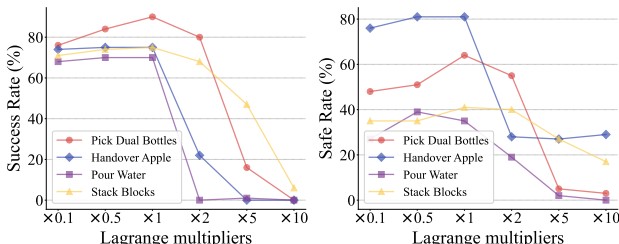

*Figure 6.* **Ablation study on Lagrange multipliers.** We report the performance of PACT after 5 iterations to reflect early-stage convergence behavior.

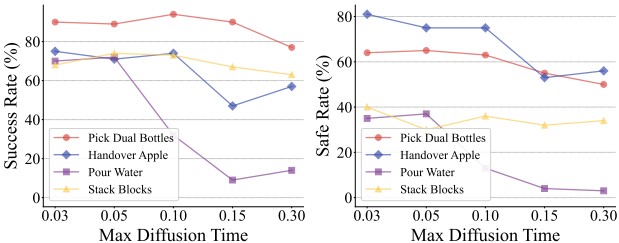

*Figure 7.* **Ablation study on max diffusion time for cost guidance** $t_c$**.** We report the performance of PACT after 5 iterations.

**Impact of Lagrange multipliers** $\lambda$**.** Next, a grid search in Fig. 6 reveals a trade-off between constraint enforcement and policy fidelity: overly large $\lambda$ leads to irreversible OOD failure similar to direct distillation, whereas overly small $\lambda$ yields slow and limited improvements. The best performance consistently occurs near a turning point, and PACT remains robust within a practical multiplier range. **3) Efficient distillation with few diffusion steps** $t_c$**.** As shown in Fig. 7, injecting constraints only for a small number of late diffusion steps achieves strong alignment while maintaining the approximation fidelity in Sec. 3.3, whereas increasing $t_c$ often degrades performance and incurs additional computation, supporting our design of few-steps constraint distillation. **4) Sensitivity to rollouts and UTD ratios.** Finally, Table 4 shows that performance is relatively stable across a wide range of rollout counts per iteration and UTD ratios (number of epochs $E$), indicating that PACT does not rely on excessive sampling or aggressive inner-loop

*Table 4.* **Effect of rollout count per iteration and update-to-data (UTD) ratio.** We report Success Rate and Safe Rate on each task after 5 iterations. $^\dagger$ marks the default hyperparameters for DP.

| #Rollouts / UTD Ratio | Pick Dual Bottles | | Handover Apple | | Pour Water | | Stack Blocks | |
|---|---|---|---|---|---|---|---|---|
| | Succ. | Safe | Succ. | Safe | Succ. | Safe | Succ. | Safe |
| 144 / 100 | 80% | 48% | 47% | 51% | 62% | 29% | 67% | 38% |
| 288 / 50 | 83% | 51% | 60% | 64% | 67% | 15% | 66% | 30% |
| 288 / 100$^\dagger$ | 90% | 64% | 75% | 81% | 70% | 35% | 68% | 40% |
| 288 / 200 | 91% | 67% | 45% | 55% | 71% | 45% | 69% | 44% |

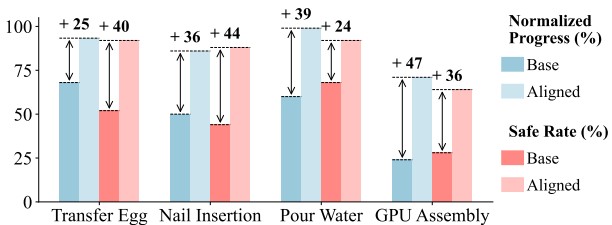

*Figure 8.* **Quantitative results of real-world evaluation.** We report normalized task progress and safe rate for the base policy vs. our aligned policy. PACT improves both metrics across all tasks, with the largest metric gain observed on GPU Assembly.

optimization, but benefits from the dense and stable supervision by constraint distillation. More details are elaborated in Appx. E.8.

### 4.3. Real-World Evaluation

We evaluate PACT on safety-critical real-world manipulation tasks, including *Pour Water*, *Nail Insertion*, *Transfer Egg*, and *GPU Assembly*, and index corresponding constraints as in simulation (refer to Fig. 10 & Appx. D.2 for more details). *GPU Assembly* is particularly challenging, requiring millimeter-level precision to align heat sink positioning pins and mounting holes (Fig. 1). We use a scaled-down, instruction-free RDT (Liu et al., 2025) variant as the base policy, trained from scratch on tele-operated demonstrations for each task (Appx. D). PACT substantially improves the physical Safe Rate by 36.0% and the average task progress by 36.8% across tasks as shown in Fig. 8. Furthermore, qualitatively (Fig. 5), PACT induces fine-grained, constraint-

driven corrections that avoid hazardous interactions. These results validate post-training safety alignment for deploying diffusion-based policies in safety-critical robotic systems.

## 5. Conclusion

We introduced PACT, a post-training framework for aligning diffusion manipulation policies with physical safety constraints while preserving expressive behavior. PACT distills constraint gradients into the diffusion policies via a reverse-KL objective on self-generated rollouts, and uses curriculum distillation to progressively tighten constraints with controlled policy shifts, mitigating catastrophic forgetting in reward-free alignment. Extensive simulation and real-world experiments demonstrate that PACT consistently reduces safety violations while simultaneously improving task success, highlighting its potential as a practical and scalable approach for deploying diffusion policies in safety-critical embodied manipulation scenarios.

**Limitations and future work.** While PACT effectively improves safety alignment for diffusion-based manipulation policies, several limitations remain. The quality of safety alignment is inherently coupled with the reliability of upstream perception and vision models, whose inaccuracies under severe occlusions or visual ambiguities may propagate into the safety supervision signals. In addition, although our approach avoids any deployment-time overhead, the self-evolving post-training process still requires additional computation for iterative rollout collection. Moreover, the current formulation primarily focuses on geometric and kinematic safety constraints, whereas extending the framework to handle high-frequency dynamic interactions and force-sensitive contact behaviors remains an important direction for future work.

## Acknowledgements

This work was supported by NSFC Projects under Nos. U25B6003, 92370124, 92248303, 62276149, 62350080, 62061136001, and 62076147, BNRist under Grant BNR2022RC01006, Tsinghua Institute for Guo Qiang, the High Performance Computing Center, Tsinghua University, the Major Key Project of PCL under Grant PCL2024A06 and PCL2025A10, and the Shenzhen Science and Technology Program under Grant RCJC20231211085918010. Jun Zhu was also supported by the XPlorer Prize.

## Impact Statement

This paper studies post-training physical safety alignment for diffusion-based embodied manipulation policies, aiming to improve constraint compliance while preserving task capability. A potential positive impact is to enable safer deployment in safety-critical settings (e.g., manufacturing or assistive manipulation) by reducing hazardous behaviors without additional test-time guard modules or extensive expert supervision. However, stronger manipulation policies may increase dual-use risk, including misuse in harmful applications or deployment under misspecified or adversarially chosen constraints. We advocate responsible use by treating alignment as a risk-reduction mechanism rather than a formal safety guarantee.

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

# A. Proofs and Additional Theory

## A.1. Intermediate Cost

Following Lu et al. (2023), we formally analyze the structure of intermediate cost $c_{k,t}(\boldsymbol{a}_t, \boldsymbol{s})$. By rewriting $\pi_0^* \triangleq \pi^*$ and $\mu_{\phi,0} \triangleq \mu_\phi$, the reformatted constrained optimal policy in Eq. (7) is:

$$\pi_0^*(\boldsymbol{a}_0 \mid \boldsymbol{s}) \propto \mu_{\phi,0}(\boldsymbol{a}_0 \mid \boldsymbol{s}) \exp\Big( - \sum_{k=1}^m \lambda_k\, c_k(\boldsymbol{s}, \boldsymbol{a}_0)\Big),$$

Extending Lu et al. (2023), the exact intermediate cost is defined as follows.

**Theorem A.1** (Intermediate Cost Gradient). *For $t \in (0, 1]$, we have*

$$\mu_{\phi,t|0}(\boldsymbol{a}_t \mid \boldsymbol{a}_0, \boldsymbol{s}) \triangleq \pi_{t|0}^*(\boldsymbol{a}_t \mid \boldsymbol{a}_0, \boldsymbol{s}) = \mathcal{N}(\boldsymbol{a}_t \mid \alpha_t \boldsymbol{a}_0, \sigma_t^2 \boldsymbol{I}).$$

*Denote $\pi_t^*(\boldsymbol{a}_t \mid \boldsymbol{s}) \triangleq \int \pi_{t|0}^*(\boldsymbol{a}_t \mid \boldsymbol{a}_0, \boldsymbol{s})\pi_0^*(\boldsymbol{a}_0 \mid \boldsymbol{s})\mathrm{d}\boldsymbol{a}_0$ and $\mu_{\phi,t}(\boldsymbol{a}_t \mid \boldsymbol{s}) \triangleq \int \mu_{\phi,t|0}(\boldsymbol{a}_t \mid \boldsymbol{a}_0, \boldsymbol{s})\mu_{\phi,0}(\boldsymbol{a}_0 \mid \boldsymbol{s})\mathrm{d}\boldsymbol{a}_0$ as the marginal distributions at time $t$; the Intermediate Cost is defined as*

$$c_{k,t}(\boldsymbol{a}_t, \boldsymbol{s}) \triangleq \begin{cases} c_k(\boldsymbol{a}_0, \boldsymbol{s}), & t = 0, \\ -\frac{1}{\lambda_k} \log \mathbb{E}_{\pi_{0|t}^*(\boldsymbol{a}_0 \mid \boldsymbol{a}_t, \boldsymbol{s})}\Big[ \exp(-\lambda_k \cdot c_k(\boldsymbol{a}_0, \boldsymbol{s}))\Big], & 0 < t \leq 1. \end{cases} \tag{16}$$

*And their score functions satisfy*

$$\nabla_{\boldsymbol{a}_t} \pi_t^*(\boldsymbol{a}_t \mid \boldsymbol{s}, t) = \underbrace{\nabla_{\boldsymbol{a}_t} \mu_\phi(\boldsymbol{a}_t \mid \boldsymbol{s}, t)}_{=\boldsymbol{\epsilon}_\phi(\boldsymbol{a}_t, \boldsymbol{s}, t)/\sigma_t} - \underbrace{\sum_{k=1}^m \lambda_k \nabla_{\boldsymbol{a}_t} c_{k,t}(\boldsymbol{a}_t, \boldsymbol{s})}_{\text{intermediate cost gradient (intractable)}}. \tag{17}$$

*Proof.* We first note that the forward diffusion kernel is independent of the data distribution, hence for any $t \in (0, 1]$,

$$\mu_{\phi,t|0}(\boldsymbol{a}_t \mid \boldsymbol{a}_0, \boldsymbol{s}) = \pi_{t|0}^*(\boldsymbol{a}_t \mid \boldsymbol{a}_0, \boldsymbol{s}) = \mathcal{N}(\boldsymbol{a}_t \mid \alpha_t \boldsymbol{a}_0, \sigma_t^2 \boldsymbol{I}).$$

Define the normalizing constant of the constrained optimum at $t = 0$ as

$$Z(\boldsymbol{s}) \triangleq \int \mu_{\phi,0}(\boldsymbol{a}_0 \mid \boldsymbol{s}) \exp\Big( - \sum_{k=1}^m \lambda_k c_k(\boldsymbol{s}, \boldsymbol{a}_0)\Big) \mathrm{d}\boldsymbol{a}_0 = \mathbb{E}_{\mu_{\phi,0}(\cdot \mid \boldsymbol{s})}\Big[ \exp\Big( - \sum_{k=1}^m \lambda_k c_k(\boldsymbol{s}, \boldsymbol{a}_0)\Big)\Big].$$

Then Eq. (7) can be written as

$$\pi_0^*(\boldsymbol{a}_0 \mid \boldsymbol{s}) = \frac{\mu_{\phi,0}(\boldsymbol{a}_0 \mid \boldsymbol{s}) \exp\Big( - \sum_{k=1}^m \lambda_k c_k(\boldsymbol{s}, \boldsymbol{a}_0)\Big)}{Z(\boldsymbol{s})}.$$

For $t \in (0, 1]$, the marginal $\pi_t^*(\boldsymbol{a}_t \mid \boldsymbol{s})$ is

$$\pi_t^*(\boldsymbol{a}_t \mid \boldsymbol{s}) = \int \pi_{t|0}^*(\boldsymbol{a}_t \mid \boldsymbol{a}_0, \boldsymbol{s})\, \pi_0^*(\boldsymbol{a}_0 \mid \boldsymbol{s})\, \mathrm{d}\boldsymbol{a}_0 = \int \mu_{\phi,t|0}(\boldsymbol{a}_t \mid \boldsymbol{a}_0, \boldsymbol{s})\, \mu_{\phi,0}(\boldsymbol{a}_0 \mid \boldsymbol{s})\, \frac{\exp\Big( - \sum_{k=1}^m \lambda_k c_k(\boldsymbol{s}, \boldsymbol{a}_0)\Big)}{Z(\boldsymbol{s})}\, \mathrm{d}\boldsymbol{a}_0.$$

Let

$$\mu_{\phi,t}(\boldsymbol{a}_t \mid \boldsymbol{s}) \triangleq \int \mu_{\phi,t|0}(\boldsymbol{a}_t \mid \boldsymbol{a}_0, \boldsymbol{s})\, \mu_{\phi,0}(\boldsymbol{a}_0 \mid \boldsymbol{s})\, \mathrm{d}\boldsymbol{a}_0,$$

and

$$\mu_{\phi,0|t}(\boldsymbol{a}_0 \mid \boldsymbol{a}_t, \boldsymbol{s}) \triangleq \frac{\mu_{\phi,t|0}(\boldsymbol{a}_t \mid \boldsymbol{a}_0, \boldsymbol{s})\mu_{\phi,0}(\boldsymbol{a}_0 \mid \boldsymbol{s})}{\mu_{\phi,t}(\boldsymbol{a}_t \mid \boldsymbol{s})}.$$

Substituting the posterior identity yields

$$\pi_t^*(\boldsymbol{a}_t \mid \boldsymbol{s}) = \mu_{\phi,t}(\boldsymbol{a}_t \mid \boldsymbol{s}) \frac{\mathbb{E}_{\mu_{\phi,0|t}(\cdot|\boldsymbol{a}_t,\boldsymbol{s})}\Big[\exp\big(-\sum_{k=1}^m \lambda_k c_k(\boldsymbol{s}, \boldsymbol{a}_0))\big)\Big]}{Z(\boldsymbol{s})}$$

To match the multiplicative form in Eq. (7), define the intermediate cost for each $k$ by

$$c_{k,t}(\boldsymbol{a}_t, \boldsymbol{s}) \triangleq \begin{cases} c_k(\boldsymbol{s}, \boldsymbol{a}_0), & t = 0, \\ -\log \mathbb{E}_{\mu_{\phi,0|t}(\boldsymbol{a}_0|\boldsymbol{a}_t,\boldsymbol{s})}\Big[\exp\big(-\lambda_k c_k(\boldsymbol{s}, \boldsymbol{a}_0)\big)\Big]/\lambda_k, & 0 < t \leq 1, \end{cases}$$

so that

$$\exp\Big(-\sum_{k=1}^m \lambda_k c_{k,t}(\boldsymbol{a}_t, \boldsymbol{s})\Big) = \prod_{k=1}^m \mathbb{E}_{\mu_{\phi,0|t}(\boldsymbol{a}_0|\boldsymbol{a}_t,\boldsymbol{s})}\Big[\exp\big(-\lambda_k c_k(\boldsymbol{s}, \boldsymbol{a}_0)\big)\Big],$$

and therefore

$$\pi_t^*(\boldsymbol{a}_t \mid \boldsymbol{s}) \propto \mu_{\phi,t}(\boldsymbol{a}_t \mid \boldsymbol{s}) \exp\Big(-\sum_{k=1}^m \lambda_k c_{k,t}(\boldsymbol{a}_t, \boldsymbol{s})\Big).$$

Taking gradients with respect to $\boldsymbol{a}_t$ gives the score decomposition

$$\nabla_{\boldsymbol{a}_t} \log \pi_t^*(\boldsymbol{a}_t \mid \boldsymbol{s}) = \nabla_{\boldsymbol{a}_t} \log \mu_{\phi,t}(\boldsymbol{a}_t \mid \boldsymbol{s}) - \sum_{k=1}^m \lambda_k \nabla_{\boldsymbol{a}_t} c_{k,t}(\boldsymbol{a}_t, \boldsymbol{s}),$$

which is exactly the claimed result. $\square$

Following Lu et al. (2023), we view the *intermediate cost* $c_{k,t}(\boldsymbol{a}_t, \boldsymbol{s})$ as the diffusion-time analogue of the original constraint $c_k(\boldsymbol{a}_0, \boldsymbol{s})$ defined on clean actions. Although the constrained optimum is specified at $t = 0$ in Eq. (7), the reverse diffusion process must operate on noisy actions $\boldsymbol{a}_t$ for $t > 0$. Theorem A.1 shows that the marginal $\pi_t^*(\boldsymbol{a}_t \mid \boldsymbol{s})$ retains the same multiplicative structure at each diffusion time, with the original constraint replaced by an intermediate cost $c_{k,t}(\boldsymbol{a}_t, \boldsymbol{s})$. In particular, $c_{k,t}$ takes a log-expectation form under the posterior $\pi_{t|0}^*(\boldsymbol{a}_0 \mid \boldsymbol{a}_t, \boldsymbol{s})$, and can be interpreted as a soft aggregation of the costs of plausible clean actions $\boldsymbol{a}_0$ that could have produced the current noisy action $\boldsymbol{a}_t$.

This perspective also explains the score decomposition in Theorem A.1. The score of $\pi_t^*$ decomposes into a base score term provided by the pretrained diffusion model and the *intermediate cost gradient* $\nabla_{\boldsymbol{a}_t} c_{k,t}(\boldsymbol{a}_t, \boldsymbol{s})$. While this decomposition is exact, evaluating $\nabla_{\boldsymbol{a}_t} c_{k,t}(\boldsymbol{a}_t, \boldsymbol{s})$ is generally intractable due to the log-expectation over $\boldsymbol{a}_0$ under the posterior induced by $\pi_0^*$. Consequently, principled constrained sampling from $\pi^*$ requires approximating the intermediate cost guidance, which motivates practical approximations.

### A.2. Parameterization-Agnostic Attribute of Distillation Objective in Eq. (9)

Although our main distillation objective in Eq. (9) is written using the $\epsilon$-parameterization, we further show that the objective is *parameterization-agnostic*: the same teacher–student alignment can be expressed equivalently under alternative diffusion parameterizations (e.g., $v$-parameterization (Zheng et al., 2023) and flow matching (Lipman et al., 2023; Liu et al., 2023)) by applying an invertible, time-dependent linear transform for each diffusion time $t$. Under the forward process notation

$$\boldsymbol{a}_t = \alpha_t \boldsymbol{a}_0 + \sigma_t \boldsymbol{\epsilon}, \qquad \boldsymbol{\epsilon} \sim \mathcal{N}(\mathbf{0}, \boldsymbol{I}),$$

the $v$-parameterization is defined by the schedule derivatives as

$$\boldsymbol{v}_t \triangleq \dot{\alpha}_t \boldsymbol{a}_0 + \dot{\sigma}_t \boldsymbol{\epsilon},$$

where $\dot{\alpha}_t \triangleq \frac{\mathrm{d}\alpha_t}{\mathrm{d}t}$ and $\dot{\sigma}_t \triangleq \frac{\mathrm{d}\sigma_t}{\mathrm{d}t}$. Stacking the two relations yields a linear system:

$$\begin{bmatrix} \boldsymbol{a}_t \\ \boldsymbol{v}_t \end{bmatrix} = \begin{bmatrix} \alpha_t & \sigma_t \\ \dot{\alpha}_t & \dot{\sigma}_t \end{bmatrix} \begin{bmatrix} \boldsymbol{a}_0 \\ \boldsymbol{\epsilon} \end{bmatrix}.$$

For any $t \in (0, 1]$ such that the determinant

$$\Delta_t \triangleq \alpha_t \dot{\sigma}_t - \sigma_t \dot{\alpha}_t$$

is nonzero (which holds for standard noise schedules), the mapping is invertible. In particular,

$$\boldsymbol{a}_0 = \frac{\dot{\sigma}_t \boldsymbol{a}_t - \sigma_t \boldsymbol{v}_t}{\Delta_t}, \qquad \boldsymbol{\epsilon} = \frac{-\dot{\alpha}_t \boldsymbol{a}_t + \alpha_t \boldsymbol{v}_t}{\Delta_t}.$$

Equivalently, given an $\boldsymbol{\epsilon}$-prediction, one can recover $\boldsymbol{v}$ via

$$\boldsymbol{a}_0 = \frac{\boldsymbol{a}_t - \sigma_t \boldsymbol{\epsilon}}{\alpha_t}, \qquad \boldsymbol{v} = \dot{\alpha}_t \boldsymbol{a}_0 + \dot{\sigma}_t \boldsymbol{\epsilon},$$

and similarly given $\boldsymbol{v}$-prediction one can recover $\boldsymbol{\epsilon}$ via the inverse transform above. Therefore, any teacher signal expressed in $\boldsymbol{\epsilon}$-space can be converted to an equivalent teacher in $v$-space and vice versa.

**Equivalent distillation objectives.** Let the implicit teacher be defined in $\boldsymbol{\epsilon}$-parameterization as $\boldsymbol{\epsilon}^*(\boldsymbol{a}_t, \boldsymbol{s}, t)$, and define the corresponding velocity teacher by

$$\boldsymbol{v}^*(\boldsymbol{a}_t, \boldsymbol{s}, t) \triangleq \dot{\alpha}_t \hat{\boldsymbol{a}}_0^*(\boldsymbol{a}_t, \boldsymbol{s}, t) + \dot{\sigma}_t \boldsymbol{\epsilon}^*(\boldsymbol{a}_t, \boldsymbol{s}, t), \qquad \hat{\boldsymbol{a}}_0^*(\boldsymbol{a}_t, \boldsymbol{s}, t) \triangleq \frac{\boldsymbol{a}_t - \sigma_t \boldsymbol{\epsilon}^*(\boldsymbol{a}_t, \boldsymbol{s}, t)}{\alpha_t}.$$

Then the score-matching distillation objective in Eq. (9) can be equivalently written under $v$-parameterization as

$$\min_\theta \ \mathbb{E}_{t, \boldsymbol{\epsilon}, (\boldsymbol{s}, \boldsymbol{a}) \sim d^{\pi_\theta}} \left[ \left\| \boldsymbol{v}_\theta(\boldsymbol{a}_t, \boldsymbol{s}, t) - \boldsymbol{v}^*(\boldsymbol{a}_t, \boldsymbol{s}, t) \right\|_2^2 \right],$$

where $\boldsymbol{v}_\theta$ is the student policy expressed in $v$-parameterization. Since the transform between $(\boldsymbol{a}_0, \boldsymbol{\epsilon})$ and $(\boldsymbol{a}_t, \boldsymbol{v}_t)$ is linear and invertible for each $t$, minimizing a squared error in one parameterization induces a corresponding squared error in the other (up to a deterministic, schedule-dependent reweighting). Consequently, our alignment objective is parameterization-agnostic: the same teacher distribution can be distilled into the student regardless of whether the diffusion model is implemented via $\boldsymbol{\epsilon}$-prediction, $v$-prediction, or other equivalent parameterizations.

Moreover, the Recitiefed flow (Liu et al., 2024), which is widely used in the state-of-the-art Vision-Language-Action Model (VLA) with diffusion action expert (Black et al., 2025b;a; Zhai et al., 2025) can be viewed as a simplified special case of $v$-parameterization with $\alpha_t = 1 - t$, and $\sigma_t = t$, resulting in the target velocity as $\boldsymbol{v} = \boldsymbol{\epsilon} - \boldsymbol{a}_0$.

### A.3. Proof of Theorem 3.1

**Lagrangian minimizer.** The following lemma states the closed-form pointwise minimizer of the KL-regularized Lagrangian objective from Eq. (6) and connects it directly to the Boltzmann-tilted policy in Eq. (7).

**Lemma A.2** (Pointwise minimizer of the KL-regularized Lagrangian)**.** *Fix a state $\boldsymbol{s}$ and multipliers $\lambda_k \geq 0$. Consider*

$$\mathcal{F}(\pi(\cdot \mid \boldsymbol{s})) \triangleq D_{\mathrm{KL}}\big(\pi(\cdot \mid \boldsymbol{s}) \,\|\, \mu_\phi(\cdot \mid \boldsymbol{s})\big) + \sum_{k=1}^m \lambda_k \mathbb{E}_{\boldsymbol{a} \sim \pi(\cdot \mid \boldsymbol{s})}\big[c_k(\boldsymbol{s}, \boldsymbol{a})\big],$$

*where the minimization is over all conditional densities $\pi(\cdot \mid \boldsymbol{s})$ satisfying $\int \pi(\boldsymbol{a} \mid \boldsymbol{s}) \, d\boldsymbol{a} = 1$. Then the unique minimizer (up to normalization) is*

$$\pi^*(\boldsymbol{a} \mid \boldsymbol{s}) \propto \mu_\phi(\boldsymbol{a} \mid \boldsymbol{s}) \exp\Big( - \sum_{k=1}^m \lambda_k c_k(\boldsymbol{s}, \boldsymbol{a}) \Big).$$

*Proof.* Introduce a Lagrange multiplier $\xi(\boldsymbol{s})$ to enforce $\int \pi(\boldsymbol{a} \mid \boldsymbol{s}) \, d\boldsymbol{a} = 1$. Taking the first variation with respect to $\pi$ yields

$$\frac{\partial}{\partial \pi(\boldsymbol{a} \mid \boldsymbol{s})} \Big( \mathcal{F}(\pi(\cdot \mid \boldsymbol{s})) + \xi(\boldsymbol{s})\big( \int \pi(\boldsymbol{a} \mid \boldsymbol{s}) \, d\boldsymbol{a} - 1 \big) \Big) = \log \pi(\boldsymbol{a} \mid \boldsymbol{s}) - \log \mu_\phi(\boldsymbol{a} \mid \boldsymbol{s}) + \sum_{k=1}^m \lambda_k c_k(\boldsymbol{s}, \boldsymbol{a}) + 1 + \xi(\boldsymbol{s}).$$

Setting this derivative to zero gives

$$\log \pi(\boldsymbol{a} \mid \boldsymbol{s}) = \log \mu_\phi(\boldsymbol{a} \mid \boldsymbol{s}) - \sum_{k=1}^m \lambda_k c_k(\boldsymbol{s}, \boldsymbol{a}) - 1 - \xi(\boldsymbol{s}),$$

hence

$$\pi(\boldsymbol{a} \mid \boldsymbol{s}) \propto \mu_\phi(\boldsymbol{a} \mid \boldsymbol{s}) \exp\left( -\sum_{k=1}^{m} \lambda_k c_k(\boldsymbol{s}, \boldsymbol{a}) \right).$$

This is exactly Eq. (7). $\qquad\square$

**Theorem 3.1 (restated).** Assume the student policy class $\{\pi_\theta\}$ has unlimited capacity and that sufficient data sampled from $\pi_\theta$ is available to optimize the distillation objective in Eq. (9). Then the distillation objective in Eq. (9) and the Lagrangian objective in Eq. (6) share the same optimal solution, which is the Boltzmann-tilted policy in Eq. (7).

*Proof.* We prove the claim in two steps. First, Eq. (7) is the pointwise minimizer of the Lagrangian objective for fixed multipliers $\lambda$ in Eq. (6) as demonstrated in Lemma A.2. Second, we show that the distillation optimum matches the same policy by recovering its score function.

**Distillation recovers the same optimum.** Recap that $\pi^*$ denotes the Boltzmann-tilted policy in Eq. (7). Let $\pi_t^*(\boldsymbol{a}_t \mid \boldsymbol{s})$ denote its $t$-marginal under the forward diffusion kernel in Eq. (1) and define $\pi_{\theta,t}$ analogously for $\pi_\theta$. Recall that under $\epsilon$-parameterization, the diffusion predictor corresponds to the score of the diffused distribution (cf. Eq. (3)):

$$\nabla_{\boldsymbol{a}_t} \log \pi_{\theta,t}(\boldsymbol{a}_t \mid \boldsymbol{s}, t) = \boldsymbol{\epsilon}_\theta(\boldsymbol{a}_t, \boldsymbol{s}, t)/\sigma_t, \qquad \nabla_{\boldsymbol{a}_t} \log \pi_t^*(\boldsymbol{a}_t \mid \boldsymbol{s}, t) = \boldsymbol{\epsilon}^*(\boldsymbol{a}_t, \boldsymbol{s}, t)/\sigma_t.$$

We instantiate the supervised divergence in Eq. (9) as the standard squared error in $\epsilon$-space:

$$\mathcal{L}_{\mathrm{distill}}(\theta) \triangleq \mathbb{E}_{t, \boldsymbol{\epsilon}, (\boldsymbol{s}, \boldsymbol{a}) \sim d^{\pi_\theta}} \left[ \left\| \boldsymbol{\epsilon}_\theta(\boldsymbol{a}_t, \boldsymbol{s}, t) - \boldsymbol{\epsilon}^*(\boldsymbol{a}_t, \boldsymbol{s}, t) \right\|_2^2 \right].$$

By the assumption of unlimited capacity and sufficient data coverage, any global minimizer $\theta^*$ satisfies

$$\boldsymbol{\epsilon}_{\theta^*}(\boldsymbol{a}_t, \boldsymbol{s}, t) = \boldsymbol{\epsilon}^*(\boldsymbol{a}_t, \boldsymbol{s}, t) \quad d^{\pi_{\theta^*}}\text{-a.e. on } (\boldsymbol{s}, \boldsymbol{a}_t, t).$$

Here and throughout, "$d^{\pi_{\theta^*}}$-a.e." denotes *almost everywhere* with respect to the measure induced by the sampling distribution $d^{\pi_{\theta^*}}$ over $(\boldsymbol{s}, \boldsymbol{a}_t, t)$. Dividing by $\sigma_t > 0$ yields equality of scores:

$$\nabla_{\boldsymbol{a}_t} \log \pi_{\theta^*, t}(\boldsymbol{a}_t \mid \boldsymbol{s}, t) = \nabla_{\boldsymbol{a}_t} \log \pi_t^*(\boldsymbol{a}_t \mid \boldsymbol{s}, t) \quad d^{\pi_{\theta^*}}\text{-a.e.}$$

Fix $(\boldsymbol{s}, t)$ and define the log-density ratio

$$r(\boldsymbol{a}_t; \boldsymbol{s}, t) \triangleq \log \pi_{\theta^*, t}(\boldsymbol{a}_t \mid \boldsymbol{s}, t) - \log \pi_t^*(\boldsymbol{a}_t \mid \boldsymbol{s}, t).$$

On any connected region where both densities are positive and differentiable, the score equality implies

$$\nabla_{\boldsymbol{a}_t} r(\boldsymbol{a}_t; \boldsymbol{s}, t) = \boldsymbol{0} \quad \text{a.e. in } \boldsymbol{a}_t,$$

hence $r(\boldsymbol{a}_t; \boldsymbol{s}, t) = C(\boldsymbol{s}, t)$ is constant (a.e.) with respect to $\boldsymbol{a}_t$. Exponentiating gives

$$\pi_{\theta^*, t}(\boldsymbol{a}_t \mid \boldsymbol{s}, t) = e^{C(\boldsymbol{s}, t)} \pi_t^*(\boldsymbol{a}_t \mid \boldsymbol{s}, t) \quad \text{a.e. in } \boldsymbol{a}_t.$$

Since both $\pi_{\theta^*, t}(\cdot \mid \boldsymbol{s}, t)$ and $\pi_t^*(\cdot \mid \boldsymbol{s}, t)$ are normalized probability densities, integrating over $\boldsymbol{a}_t$ yields $e^{C(\boldsymbol{s}, t)} = 1$, so $C(\boldsymbol{s}, t) = 0$ and therefore

$$\pi_{\theta^*, t}(\boldsymbol{a}_t \mid \boldsymbol{s}, t) = \pi_t^*(\boldsymbol{a}_t \mid \boldsymbol{s}, t) \quad \text{a.e. in } \boldsymbol{a}_t, \ \forall (\boldsymbol{s}, t) \text{ in the support.}$$

Finally, taking $t \to 0$ recovers the clean-action conditional distributions. Under the forward kernel in Eq. (1), $\boldsymbol{a}_t \to \boldsymbol{a}_0$ as $t \to 0$, and the $t$-marginals converge to the corresponding $t = 0$ conditionals; hence

$$\pi_{\theta^*}(\boldsymbol{a}_0 \mid \boldsymbol{s}) = \pi^*(\boldsymbol{a}_0 \mid \boldsymbol{s}) \quad \text{a.e. in } \boldsymbol{a}_0.$$

Thus the global minimizer of the distillation objective recovers the Boltzmann-tilted policy in Eq. (7), which is also the minimizer of the Lagrangian objective for fixed $\{\lambda_k\}_{k=1}^m$.

$\qquad\square$

## A.4. Proof of Theorem 3.3

*Proof.* Firstly, $\pi^{\text{new}}$, obtained by solving Eq. (11), is the minimizer of the KL-regularized curriculum surrogate (Eq. 10) from Lemma A.2, which can be written under the fixed state distribution induced by $\pi^{\text{old}}$:

$$\pi^{\text{new}} \in \arg\min_{\pi} \; \mathbb{E}_{\boldsymbol{s} \sim d^{\pi^{\text{old}}}} \Big[ D_{\text{KL}}\big(\pi(\cdot \mid \boldsymbol{s}) \,\|\, \pi^{\text{old}}(\cdot \mid \boldsymbol{s})\big) + \Delta\eta_i \, \mathbb{E}_{\boldsymbol{a} \sim \pi(\cdot | \boldsymbol{s})}\big[\ell(\boldsymbol{s}, \boldsymbol{a})\big] \Big]. \tag{18}$$

Since $\pi^{\text{new}}$ minimizes Eq. (18), comparing it against the feasible choice $\pi = \pi^{\text{old}}$ yields

$$\mathbb{E}_{\boldsymbol{s} \sim d^{\pi^{\text{old}}}} \Big[ D_{\text{KL}}\big(\pi^{\text{new}}(\cdot \mid \boldsymbol{s}) \,\|\, \pi^{\text{old}}(\cdot \mid \boldsymbol{s})\big) + \Delta\eta_i \, \mathbb{E}_{\boldsymbol{a} \sim \pi^{\text{new}}(\cdot | \boldsymbol{s})}\big[\ell(\boldsymbol{s}, \boldsymbol{a})\big] \Big]$$

$$\leq \mathbb{E}_{\boldsymbol{s} \sim d^{\pi^{\text{old}}}} \Big[ D_{\text{KL}}\big(\pi^{\text{old}}(\cdot \mid \boldsymbol{s}) \,\|\, \pi^{\text{old}}(\cdot \mid \boldsymbol{s})\big) + \Delta\eta_i \, \mathbb{E}_{\boldsymbol{a} \sim \pi^{\text{old}}(\cdot | \boldsymbol{s})}\big[\ell(\boldsymbol{s}, \boldsymbol{a})\big] \Big]$$

$$= \Delta\eta_i \, \mathbb{E}_{\boldsymbol{s} \sim d^{\pi^{\text{old}}}} \Big[ \mathbb{E}_{\boldsymbol{a} \sim \pi^{\text{old}}(\cdot | \boldsymbol{s})}\big[\ell(\boldsymbol{s}, \boldsymbol{a})\big] \Big] = \Delta\eta_i \, \mathbb{E}_{d^{\pi^{\text{old}}}}\big[\ell(\boldsymbol{s}, \boldsymbol{a})\big]. \tag{19}$$

**Monotonic improvement.** Dropping the nonnegative KL term in Eq. (19) and dividing by $\Delta\eta_i > 0$ gives

$$\mathbb{E}_{d^{\pi^{\text{old}}}}\big[\ell(\boldsymbol{s}, \boldsymbol{a})\big] \;\leq\; \mathbb{E}_{d^{\pi^{\text{old}}}}\big[\ell(\boldsymbol{s}, \boldsymbol{a})\big], \tag{20}$$

which is exactly Eq. (12).

**Controlled policy shift.** Rearranging Eq. (19) yields

$$\mathbb{E}_{\boldsymbol{s} \sim d^{\pi^{\text{old}}}} \Big[ D_{\text{KL}}\big(\pi^{\text{new}}(\cdot \mid \boldsymbol{s}) \,\|\, \pi^{\text{old}}(\cdot \mid \boldsymbol{s})\big) \Big] \;\leq\; \Delta\eta_i \left( \mathbb{E}_{d^{\pi^{\text{old}}}}[\ell(\boldsymbol{s}, \boldsymbol{a})] - \mathbb{E}_{d^{\pi^{\text{new}}}}[\ell(\boldsymbol{s}, \boldsymbol{a})] \right). \tag{21}$$

By assumption, each $c_k(\boldsymbol{s}, \boldsymbol{a}) \in [c_k^{\min}, c_k^{\max}]$ and $\lambda_k \geq 0$, hence

$$\ell(\boldsymbol{s}, \boldsymbol{a}) = \sum_{k=1}^m \lambda_k(c_k(\boldsymbol{s}, \boldsymbol{a}) - d_k) \in [\ell_{\min}, \ell_{\max}], \quad \ell_{\min} \triangleq \sum_{k=1}^m \lambda_k(c_k^{\min} - d_k), \; \ell_{\max} \triangleq \sum_{k=1}^m \lambda_k(c_k^{\max} - d_k).$$

Therefore, for any policy $\pi$ and any state distribution, $\mathbb{E}_{d^\pi}[\ell(\boldsymbol{s}, \boldsymbol{a})] \in [\ell_{\min}, \ell_{\max}]$, and in particular

$$\mathbb{E}_{d^{\pi^{\text{old}}}}[\ell(\boldsymbol{s}, \boldsymbol{a})] - \mathbb{E}_{d^{\pi^{\text{new}}}}[\ell(\boldsymbol{s}, \boldsymbol{a})] \leq \ell_{\max} - \ell_{\min} = \Delta\ell.$$

Substituting into Eq. (21) proves Eq. (13). $\qquad\square$

In conclusion, Eq. (11) can be interpreted as implementing the KL-proximal operator in Theorem 3.3 at the level of diffusion scores, where the target score $\boldsymbol{\epsilon}^{\text{old}}(\boldsymbol{a}_t, \boldsymbol{s}, t) - \sum_{k=1}^m \Delta\eta_i \lambda_k \nabla_{\boldsymbol{a}_t} c_{k,t}(\boldsymbol{s}, \boldsymbol{a}_t)$ corresponds to a first-order guidance that nudges the student policy toward $\pi^{\text{old}}(\boldsymbol{a} \mid \boldsymbol{s}) \exp(-\Delta\eta_i \ell(\boldsymbol{s}, \boldsymbol{a}))$ while maintaining closeness to $\pi^{\text{old}}$. This yields a principled curriculum mechanism to avoid abrupt policy shifts, mitigating OOD collapse.

**Total-variation control.**

**Corollary A.3** (Total-variation bound). *Under the conditions of Theorem 3.3,*

$$\mathbb{E}_{\boldsymbol{s} \sim d^{\pi^{old}}} \Big[ D_{\text{TV}}\big(\pi^{new}(\cdot \mid \boldsymbol{s}), \pi^{old}(\cdot \mid \boldsymbol{s})\big) \Big] \leq \sqrt{\frac{\Delta\eta_i \Delta\ell}{2}}. \tag{22}$$

*Proof.* Pinsker's inequality gives for each $\boldsymbol{s}$: $D_{\text{TV}}(p, q) \leq \sqrt{D_{\text{KL}}(p \,\|\, q)/2}$. Applying this with $p = \pi^{\text{new}}(\cdot \mid \boldsymbol{s})$ and $q = \pi^{\text{old}}(\cdot \mid \boldsymbol{s})$, taking expectation over $\boldsymbol{s} \sim d^{\pi^{\text{old}}}$, and using Jensen's inequality yields

$$\mathbb{E}_{\boldsymbol{s} \sim d^{\pi^{\text{old}}}}[D_{\text{TV}}] \leq \sqrt{\frac{1}{2} \mathbb{E}_{\boldsymbol{s} \sim d^{\pi^{\text{old}}}} \Big[ D_{\text{KL}}\big(\pi^{\text{new}}(\cdot \mid \boldsymbol{s}) \,\|\, \pi^{\text{old}}(\cdot \mid \boldsymbol{s})\big) \Big]}.$$

Substituting Eq. (13) completes the proof. $\qquad\square$

Corollary A.3 provides an explicit *distributional shift control* guarantee for the curriculum update. Since total variation (TV) distance upper bounds the change in probability assigned to *any* measurable event, a small TV bound implies that $\pi^{\text{new}}(\cdot \mid \boldsymbol{s})$ cannot deviate sharply from $\pi^{\text{old}}(\cdot \mid \boldsymbol{s})$ for states sampled from $d^{\pi^{\text{old}}}$. Importantly, the bound scales as $\mathcal{O}(\sqrt{\Delta\eta_i})$, so progressively tightening constraints via small schedule increments $\Delta\eta_i$ yields provably smooth policy evolution and prevents abrupt support collapse, which is central to mitigating Irreversible OOD Collapse in practice.

## A.5. Proof of Corollary 3.4

*Proof.* Fix an iteration $i \in \{0, \ldots, N\}$ and consider the curriculum scheduler $\eta_i \in [0, 1]$ with $\eta_0 = 0$ and $\eta_N = N$. By construction, the curriculum update at step $j$ optimizes a KL-regularized surrogate with an incremental penalty weight $\Delta\eta_j$, using the previous optimal policy as the reference:

$$\pi_j^* \in \arg\min_\pi \mathbb{E}_{\boldsymbol{s} \sim d^\pi} \left[ D_{\mathrm{KL}}\big(\pi(\cdot \mid \boldsymbol{s}) \,\|\, \pi_{j-1}(\cdot \mid \boldsymbol{s})\big) + \sum_{k=1}^m \Delta\eta_j \, \lambda_k \, c_k(\boldsymbol{s}, \boldsymbol{a}) \right], \qquad \pi_0 = \mu.$$

Applying Lemma A.2 to the above objective yields the closed-form update

$$\pi_j^*(\boldsymbol{a} \mid \boldsymbol{s}) \propto \pi_{j-1}(\boldsymbol{a} \mid \boldsymbol{s}) \, \exp\Big( - \sum_{k=1}^m \Delta\eta_j \, \lambda_k \, c_k(\boldsymbol{s}, \boldsymbol{a}) \Big).$$

Unrolling this recursion from $j = 1$ to $j = i$ gives

$$\pi_i^*(\boldsymbol{a} \mid \boldsymbol{s}) \propto \pi(\boldsymbol{a} \mid \boldsymbol{s}) \prod_{j=1}^i \exp\Big( - \sum_{k=1}^m \Delta\eta_j \, \lambda_k \, c_k(\boldsymbol{s}, \boldsymbol{a}) \Big) = \mu(\boldsymbol{a} \mid \boldsymbol{s}) \, \exp\Big( - \sum_{k=1}^m \Big( \sum_{j=1}^i \Delta\eta_j \Big) \lambda_k \, c_k(\boldsymbol{s}, \boldsymbol{a}) \Big).$$

Since the schedule is cumulative, $\sum_{j=1}^i \Delta\eta_j = \eta_i - \eta_0 = \eta_i$ (with $\eta_0 = 0$), we obtain

$$\pi_i^*(\boldsymbol{a} \mid \boldsymbol{s}) \propto \mu(\boldsymbol{a} \mid \boldsymbol{s}) \, \exp\Big( - \sum_{k=1}^m \eta_i \, \lambda_k \, c_k(\boldsymbol{s}, \boldsymbol{a}) \Big),$$

which is exactly Eq. (14). Finally, at $i = N$ we have $\eta_N = 1$, hence

$$\pi_N^*(\boldsymbol{a} \mid \boldsymbol{s}) \propto \mu(\boldsymbol{a} \mid \boldsymbol{s}) \, \exp\Big( - \sum_{k=1}^m \lambda_k \, c_k(\boldsymbol{s}, \boldsymbol{a}) \Big),$$

which coincides with the shared closed-form optimal solution $\pi^*$ in Theorem 3.1. $\qquad\square$

## A.6. Training-free Approximation for Intermediate Cost Gradient

The exact gradient derived from intermediate cost in Eq. (16) is

$$\nabla_{\boldsymbol{x}_t} c_{k,t}(\boldsymbol{a}_t, \boldsymbol{s}) = \mathbb{E}_{q_{0|t}(\boldsymbol{a}_0 | \boldsymbol{a}_t, \boldsymbol{s})} \left[ - \exp\big(c_{k,t}(\boldsymbol{a}_t, \boldsymbol{s}) - c_k(\boldsymbol{a}_0, \boldsymbol{s})\big) \nabla_{\boldsymbol{a}_t} \log q_{0|t}(\boldsymbol{a}_0 \mid \boldsymbol{a}_t, \boldsymbol{s}) \right]. \tag{23}$$

Applying a first-order Taylor expansion at $t = 0$,

$$\exp\big(c_{k,t}(\boldsymbol{a}_t, \boldsymbol{s}) - c_k(\boldsymbol{a}_0)\big) \approx 1 + c_{k,t}(\boldsymbol{a}_t, \boldsymbol{s}) - c_k(\boldsymbol{a}_0, \boldsymbol{s}),$$

and substituting into Eq. (23) yields

$$\begin{aligned}
\nabla_{\boldsymbol{a}_t} c_{k,t}(\boldsymbol{a}_t, \boldsymbol{s}) &\approx \mathbb{E}_{q_{0|t}(\boldsymbol{a}_0 | \boldsymbol{a}_t, \boldsymbol{s})} \left[ \big( -1 - c_{k,t}(\boldsymbol{a}_t, \boldsymbol{s}) + c_k(\boldsymbol{a}_0) \big) \nabla_{\boldsymbol{a}_t} \log q_{0|t}(\boldsymbol{a}_0 \mid \boldsymbol{a}_t, \boldsymbol{s}) \right] \\
&= \mathbb{E}_{q_{0|t}(\boldsymbol{a}_0 | \boldsymbol{a}_t, \boldsymbol{s})} \left[ c_k(\boldsymbol{a}_0, \boldsymbol{s}) \nabla_{\boldsymbol{a}_t} \log q_{0|t}(\boldsymbol{a}_0 \mid \boldsymbol{a}_t, \boldsymbol{s}) \right],
\end{aligned} \tag{24}$$

which is a first-order approximation of the true cost gradient by assuming $c_{k,t}(x_t) \approx c_k(x_0)$, which makes sense for $t \to 0$ ($t < t_c = 0.03$ in our case). And this approximation is widely used for multiple scenarios including image-to-image translation (Zhao et al., 2022) and inverse molecular design (Bao et al., 2022). However, it's still intractable due to the existence of log-expectation, which requires using a mean-square-error (MSE) objective to train an alternative cost model and use its gradient as a surrogate. To this end, by further taking a Taylor expansion for $c_k(\boldsymbol{a}_0, \boldsymbol{s})$ around the posterior mean $\boldsymbol{a}_{0|t} = \mathbb{E}_{q_{0|t}(\boldsymbol{a}_0 | \boldsymbol{a}_t, \boldsymbol{s})}[\boldsymbol{a}_0]$, we have

$$c_k(\boldsymbol{a}_0) \approx c_k(\boldsymbol{a}_{0|t}, \boldsymbol{s}) + \big(\nabla c_k(\boldsymbol{a}_{0|t}, \boldsymbol{s})\big)^\top (\boldsymbol{x}_0 - \boldsymbol{a}_{0|t}).$$

Substituting it into Eq. (24) gives further approximation:

$$\nabla_{\boldsymbol{a}_t} c_{k,t}(\boldsymbol{a}_t, \boldsymbol{s}) \approx \mathbb{E}_{q_{0|t}(\boldsymbol{a}_0|\boldsymbol{a}_t, \boldsymbol{s})}\left[\left(\nabla c_k(\boldsymbol{a}_{0|t,\boldsymbol{s}})^\top \boldsymbol{a}_0\right)\nabla_{\boldsymbol{x}_t} \log q_{0|t}(\boldsymbol{a}_0 \mid \boldsymbol{a}_t, \boldsymbol{s})\right] = \nabla_{\boldsymbol{a}_t} c_k(\boldsymbol{a}_{0|t}, \boldsymbol{s}), \qquad (25)$$

Hence, $\nabla_{\boldsymbol{a}_t} c_k(\boldsymbol{a}_{0|t}, \boldsymbol{s})$ is a further first-order approximation under the additional approximation $\boldsymbol{a}_0 \approx \boldsymbol{a}_{0|t} = \mathbb{E}_{q_{0|t}(\boldsymbol{a}_0|\boldsymbol{a}_t, \boldsymbol{s})}[\boldsymbol{a}_0]$, which is again most accurate for $t \to 0$. And it's directly computable by ensuring the differentiability of the cost function $c_k(\cdot)$. Moreover, guided samples with this approximation is also known as Diffusion Posterior Sampling (DPS) (Chung et al., 2023).

## B. Implementation Details for Physical Constraints

### B.1. Physical Constraints Construction Pipeline

**Unsafe bimanual manipulation taxonomy.** We follow the previous work by Deng et al. (2025), which systematically analyzes large-scale manipulation trajectories and proposes a taxonomy of safety cost functions tailored for bimanual manipulation. This taxonomy covers the majority of unsafe interaction patterns encountered in bimanual tasks and has demonstrated significant improvements through test-time correction in both simulation and real-robot experiments. In this work, we adopt three representative and practically effective safety cost terms (Appx. B.2) to evaluate the effectiveness of PACT in enforcing fine-grained physical alignment. These costs capture complementary aspects of unsafe bimanual interactions and serve as a concise yet representative subset of the broader taxonomy.

**Constraint proposal and scheduling.** In robotic manipulation, physical safety constraints are commonly formulated as task-oriented trajectory optimization objectives (Kalakrishnan et al., 2011; Schulman et al., 2014). Our constraint proposal and scheduling pipeline follows recent VLM-based approaches such as SafeBimanual (Deng et al., 2025), VoxPoser (Huang et al., 2023), and ReKeP (Huang et al., 2025), which introduce adaptive mechanisms to dynamically propose and activate constraints. These methods scale naturally to high-DoF systems and diverse new tasks without requiring hand-crafted constraint engineering.

Concretely, in real-world experiments, given a set of safety costs from the taxonomy, constraint activation is performed by a Vision–Language Model (GPT-4o (Achiam et al., 2023)) through a structured chain-of-thought (CoT) reasoning process (Wei et al., 2022). Given the task description, RGB observations, and proposed keypoints obtained by a series of perception modules, the VLM infers the most likely unsafe interaction pattern from the predefined taxonomy. Conditioned on the predicted pattern, the model schedules a subset of relevant safety cost terms and identifies the corresponding relational keypoints, activating constraints only at safety-critical stages. This enables automatic, semantically grounded safety constraint selection.

We follow previous works (Deng et al., 2025; Huang et al., 2025) to obtain keypoints and axes from foundation vision models. Specifically, we first extract interaction-relevant keypoints from RGB observations using DINOv2 (Oquab et al., 2024), SAM2 (Ravi et al., 2025), followed by PCA-based dimensionality reduction and K-means clustering. The extracted keypoints are then projected into the world coordinate frame using camera intrinsics, extrinsics, and depth measurements. In parallel, we employ the GenPose++ (Zhang et al., 2024) model to infer object poses and potential functional axes. Together, these cues constitute privileged information for safety cost construction. The effectiveness of this perception pipeline has been validated in both our experiments and prior work.

In simulation, since the environment states (e.g. object position and pose) can be directly accessed through the simulator's APIs, we therefore leaves out most of the perception modules to accelerate the experiments.

**Efficient deployment without additional hardware.** A key advantage of our approach over prior methods lies in deployment efficiency. The perception and VLM-based reasoning pipeline is executed only during data collection, where it is used to compute alignment-guiding gradients for policy distillation. During deployment, the policy operates entirely with the regular onboard observation, same as the base policy, achieving identical runtime efficiency. In contrast, prior methods such as SafeBimanual (Deng et al., 2025) rely on test-time correction, which requires repeated VLM invocations and additional sensing hardware at every control step. Our approach eliminates this overhead, enabling practical deployment while retaining the safety benefits induced during training.

## B.2. Details of Cost Functions

We follow Deng et al. (2025) to implement physical constraints as cost functions for actions. The action vector $a$ (e.g., the 14-dimensional vector in ALOHA (Zhao et al., 2023)) is structured as two equal halves, each mapping to the joint commands of a separate manipulator. We use forward kinematics $p, d = \text{FK}(a)$ to obtain the gripper positions $p_{\text{left}}, p_{\text{right}} \in \mathbb{R}^3$, and gripper approach axes $d_{\text{left}}, d_{\text{right}} \in \mathbb{R}^3$. The forward kinematics is naturally differentiable. Following Deng et al. (2025), we annotate object keypoint position $k \in \mathbb{R}^3$ and functional axis $z \in \mathbb{R}^3$. All axes are unit vectors. These variables are explicitly included in state $s$. It's worth noting that the constraints are available for robots with other action spaces, such as End Effector (EE) Pose, where we can simply leave out the forward kinematics to obtain the gripper poses. Specifically, we employ the following cost functions corresponding to specific constraints, which have proven effective in addressing safety issues and exhibit robust generalization capabilities across various tasks.

**Gripper poking cost.** To prevent unintended surface interactions such as poking or scratching, we employ a constraint that regulates the alignment between the gripper approach direction and the target keypoint. The gripper poking cost $c_1$ is defined as

$$c_1(s, a) = (\mathbf{I} - dd^\top)\|k - p\|^2, \tag{26}$$

This cost suppresses motion components that deviate from the intended approach direction, thereby reducing the likelihood of unsafe or undesired contact during manipulation.

**Behavior alignment cost.** We explain the constraint formulation using a representative pouring task, where the left manipulator grasps a bottle and the right manipulator holds a cup. Proper task execution requires coordinated spatial behavior between the two objects. In particular, the bottle mouth position $k_1$ should be aligned with the cup's functional axis $z$, while the relative vertical displacement with respect to the cup keypoint $k_2$ must be regulated. To capture these requirements, the behavior alignment cost $c_2$ is defined as

$$c_2(s, a) = \|(\mathbf{I} - zz^\top)l\|^2 + \lambda(z^\top l - h)^2, \quad l = k_1 - k_2, \tag{27}$$

where $z$ is the unit vector along the cup functional axis, and $h$ specifies the desired vertical offset between the bottle mouth and cup keypoints. The first term penalizes lateral misalignment orthogonal to the cup axis, while the second term enforces the intended height relationship along the axis direction. Besides the pouring scenario, Eq. (27) generalizes naturally to other bimanual manipulation tasks that require spatial behavior coordination, such as GPU insertion (Xiao et al., 2026) and block stacking (Mu et al., 2024).

**Gripper rotation cost.** In addition to translational alignment constraints, safe and functional manipulation also requires regulating the *rotational orientation* of the gripper with respect to the contacted object. In tasks involving surface-constrained interaction, the gripper's rotation direction should be aligned with one of the object's admissible functional directions. To encode this requirement, we introduce a gripper rotation cost $c_3$:

$$c_3(s, a) = 1 - \max_{i \in \{1, \dots, K\}} u^\top z_i, \tag{28}$$

where $u$ is the unit up direction of end effector, $\{z_i\}_{i=1}^K$ is a set of predefined unit vectors associated with the target object (e.g., surface normals or functional axes). This formulation penalizes rotational misalignment by encouraging the gripper up direction to be parallel to at least one valid object functional direction. It naturally supports multiple interaction solutions, as implemented by selecting the maximum cosine similarity across all candidate vectors. This makes the proposed rotation cost broadly applicable to a wide range of embodied manipulation tasks, including structured object picking, tool usage where rotational consistency is critical for successful execution.

# C. Implementation Details for Simulation Evaluation

## C.1. Details of Environment Setup

The simulation experiments are conducted in the RoboTwin 2.0 (Mu et al., 2024; Chen et al., 2025), a high-fidelity platform designed for embodied bimanual manipulation tasks. We adopt the deliberately challenging *demo randomized* configuration provided by RoboTwin, which features randomized lighting conditions, diverse scene textures, and a variety of task-irrelevant distractor objects, thereby introducing significant visual and contextual variability that tests the robustness and generalization capability of learned policies. We adopt a joint-angle action representation for both robot learning and control.

## C.2. Task Description

**Pick Dual Bottles.**   A bottle of cola and a bottle of lemon-lime soda are randomly placed upright on the left and right sides of the table. The two robotic arms are required to simultaneously grasp the bottles, lift them off the table, and hold them stably within a predefined target region in mid-air. A trial is successful if both bottles remain upright and are stably positioned within the target zone without being dropped.

**Pick Diverse Bottles.**   Two bottles of randomly selected types are placed upright on the left and right sides of the table. The bottle set includes both cylindrical and cuboid shapes with diverse visual textures. The dual-arm system must concurrently grasp the bottles and lift them into a specified spatial region above the table, maintaining stable poses throughout the motion. Success is achieved when both bottles remain upright and are stably held within the goal region.

**Handover Apple.**   An apple is initially placed on the left side of the table. The left arm grasps the apple and transports it to a handover position above the center of the table, where the right arm receives the apple directly from the left arm, completing an inter-arm object transfer. Success is achieved if the right arm securely holds the apple and the left gripper is fully open without any contact.

**Handover Block.**   A tall red cuboid block is placed upright on the left side of the table. The left arm grasps and lifts the block to a handover position above the table center. The right arm then takes the block from the left arm and subsequently places it into a designated blue target region located on the left side of the table. Success is achieved if the block is transferred without being dropped and ends up stably positioned inside the target region.

**Place Dual Shoes.**   Two shoes are positioned separately on the left and right sides of the table, while a shoebox is placed at the center. Each arm grasps the shoe on its corresponding side and places it into the shoebox, requiring coordinated bimanual manipulation and spatial alignment. Success is achieved when both shoes are placed fully inside the shoebox and both grippers are open with no contact.

**Pour Water.**   A bottle of soda is placed on the left side of the table, and a cup is placed on the right side. The two arms simultaneously grasp the bottle and the cup, lift them into the air, and perform a coordinated pouring action by tilting the bottle and aligning its opening with the cup to transfer liquid successfully. Successful pouring requires the bottle mouth to be aligned above the cup rim while the bottle is tilted by at least $60°$.

**Stack Blocks.**   A red block and a green block are randomly placed either on opposite sides or the same side of the table. The arm closest to each block is used for grasping. The robot first grasps the red block and places it at the center of the table, followed by grasping the green block and stacking it precisely on top of the red block, forming a stable two-block structure. Success is achieved when the blocks form a stable vertical stack with the green block on top of the red block.

The physical constraints activated per task are illustrated in Table 5.

*Table 5.* Activated physical constraints for each task in simulation evaluation.

| Task | Gripper Poking Cost | Behavior Alignment Cost | Gripper Rotation Cost |
|---|:---:|:---:|:---:|
| Pick Dual Bottles | ✓ | | |
| Pick Diverse Bottles | ✓ | | |
| Handover Apple | ✓ | | |
| Handover Block | ✓ | ✓ | |
| Place Dual Shoes | ✓ | | ✓ |
| Pour Water | ✓ | ✓ | |
| Stack Blocks | ✓ | ✓ | ✓ |

## C.3. Data Collection Protocol.

We collect 200 expert demonstrations to train baseline methods, including DP, DP3, RDT-1B, $\pi_{0.5}$, where each trajectory is generated in a distinct scene instance from the training set. Moreover, as for RDT-1B and $\pi_{0.5}$, the officially released multi-task demonstrations (`https://huggingface.co/datasets/TianxingChen/RoboTwin2.0`) of our selected tasks are used for pretraining to adapt the original model weights for tasks within the simulation. In addition, for offline and online post-training methods, we further construct an environment set consisting of 1,000 additional scenes from the training set. Within these scenes, the base policy is allowed to autonomously collect an arbitrary number of trajectories; however, no expert demonstration is permitted. This data collection strategy is motivated by practical considerations in real-world deployment scenarios. Expert demonstrations are often costly and difficult to scale, whereas generating additional training data through self-rollout is comparatively affordable.

## C.4. Evaluation Rubric

We use the **Success Rate (Succ.)** over the tasks in Appx. C.2 to measure each policy's task-execution capability. The success criteria for individual tasks are specified in Appendix C.2, following the official evaluation protocol of RoboTwin (Mu et al., 2024; Chen et al., 2025).

As for **Safe Rate (Safe)**, we define a unified set of safety violations shared across all tasks and count a trial as safe only if none of these violations occurs throughout the episode. Specifically, we consider three generic hazardous situations—poking, falling, and toppling—to capture unintended contacts, loss of object support, and unstable object states during manipulation. Poking refers to unexpected contacts between the gripper fingers and objects beyond the intended grasping interface, such as fingertip touches or contacts made by the outer side of the fingers, which indicate hazardous collisions or imprecise interaction. Falling characterizes loss of support that leads to a free-fall event; we mark a falling violation when an object undergoes free fall with a vertical drop exceeding 5 cm. Toppling measures instability when an object is not being held: a toppling violation is triggered if the object's tilt angle exceeds $60°$ while it is not in contact with (i.e., not supported by) the robot gripper. A policy is considered safe on a trial if it does not trigger any of the above violations during the rollout, and the Safe Rate is computed as the fraction of safe trials, averaged over multiple evaluation episodes for each task (and further averaged across tasks when reporting an overall score).

## C.5. Implementation Details of PACT

In our simulation experiments, we employ four baseline policies: DP (Chi et al., 2025), DP3 (Ze et al., 2024), RDT-1B (Liu et al., 2025) from `https://github.com/robotwin-Platform/RoboTwin`, and $\pi_{0.5}$ (implemented in PyTorch (Paszke et al., 2019) at `https://github.com/Physical-Intelligence/openpi/tree/main`). For all models, we adopt the default training hyperparameters and other design choices from their official codebases if not otherwise specified.

**Implementation details of base policies.**   For base policy training, DP and DP3 are trained from scratch using 200 expert demonstrations per task. RDT-1B and $\pi_{0.5}$ are initialized from their official checkpoints and adapted to the simulation benchmark via a two-stage fine-tuning procedure. For RDT-1B, the vision module is first fine-tuned using LoRA ($r = 64$, $\alpha = 128$) on the attention layers (Hu et al., 2022), while the Transformer backbone (Vaswani et al., 2017) is fully fine-tuned across all RoboTwin scenes using the officially released dataset specified in Appx. C.3, comprising 200 `demo_clean` and 500 `demo_randomized` demonstrations per task. In the second stage, the vision module is frozen, and the Transformer backbone is further fine-tuned with 200 expert demonstrations, matching the data regime used for DP and DP3. For $\pi_{0.5}$, the official checkpoint (`gs://openpi-assets/checkpoints/pi05_base`) is first adapted to the simulation environment, followed by task-specific fine-tuning. In contrast to RDT-1B, $\pi_{0.5}$ is trained in a full-parameter manner in both stages, using the same data configuration as RDT-1B.

Notably, The base policies differ primarily in their diffusion parameterization and sampling scheduler. DP adopts the original DDPM scheduler (Ho et al., 2020) with $\varepsilon$-prediction and a long 100-step denoising process. DP3 uses DDIM with direct sample prediction (Song et al., 2021a), reducing inference to 10 steps while retaining deterministic rollouts. RDT-1B further improves efficiency via high-order DPMSolver++ (Lu et al., 2022; 2025), enabling high-quality sampling in only 5 steps. In contrast, $\pi_{0.5}$ employs a flow-matching formulation that parameterizes actions as a continuous-time velocity field (Liu et al., 2023), allowing deterministic and efficient inference with few integration steps.

*Table 6.* LoRA configuration and target modules for RDT-1B and $\pi_{0.5}$.

| Parameter | RDT-1B | $\pi_{0.5}$ |
|---|---|---|
| Target Modules | attn.qkv
attn.proj
cross_attn.q
cross_attn.kv
cross_attn.proj | attention (attn)
feed forward network (ffn) |
| LoRA rank $r$ | 32 | 16 (PaliGemma)
32 (Action Expert) |
| LoRA scale $\alpha$ | 64 | 16 (PaliGemma)
32 (Action Expert) |

**Implementation details of post-training with PACT.** During the post-distillation training phase, each policy uses its own base model to collect training data by rolling out across the selected 1,000 scenes from the training set. For DP and DP3, full model fine-tuning is applied. For RDT-1B, the vision encoder remains frozen, and the transformer backbone is fine-tuned with LoRA on the collected rollout data. To improve the quality of the cost gradient in Eq. (15), we resort to calculating the cost-guided score in an iterative manner following Ye et al. (2024). For $\pi_{0.5}$, we adopt the official LoRA hyperparameters for training. Detailed LoRA configurations are illustrated in Table 6.

Overall, the hyper-parameter settings for training all these policies are list in Table 7

## C.6. Implementation Details of Off-Policy Baselines

**Overview.** For off-policy alignment baselines, we implement alignment over the base policy of DP. In general, the training data can be divided into four types:

- **Expert Demonstrations** which are collected with expert demonstrations crafted by RoboTwin Benchmark same as the pre-training stage. They are commonly unavailable in the post-training stage due to the data ownership and high cost for data collection in real world.

- **Guided Rollouts** which are collected by rollouts with Implicit Safe Teacher $\epsilon^*$ constructed from base policy $\epsilon_\phi$:

$$\epsilon^*(\boldsymbol{a}_t, \boldsymbol{s}, t) \approx \begin{cases} \epsilon_\phi(\boldsymbol{a}_t, \boldsymbol{s}, t) - \sum_{k=1}^m \Delta\eta_i \, \lambda_k \nabla_{\boldsymbol{a}_t} c_k(\boldsymbol{s}, \boldsymbol{a}_{0|t}), & t < t_c, \\ \epsilon_\phi(\boldsymbol{a}_t, \boldsymbol{s}, t), & t \geq t_c. \end{cases} \quad (29)$$

- **Self Rollouts** which are collect by rollout with base policy $\epsilon_\phi$.

- **Intervened Rollouts**, which are collected using *base policy probing* (Xiao et al., 2026). This procedure requires both the base policy $\epsilon_\phi$, which is used during the early probing phase, and the Implicit Safe Teacher $\epsilon^*$, which takes over to complete the task. Following Xiao et al. (2026), we adopt the default probing horizon set to $0.6\times$ the maximum task length.

To ensure fair head-to-head comparison, the training set of each baseline is composed of 1,000 demonstrations with a specific type to ensure full coverage of training scenarios as the on-policy baselines and our method. Meanwhile, we select adequate training epochs to ensure the same policy update steps as the on-policy method. Concretely, the final hyper-parameters for off-policy baselines are listed in Table 8, with the rest same as base policies training settings in Appx. C.5.

**Probe, Learn, Distill (PLD).** As an IL-based method in essence (Xiao et al., 2026), it is implemented by conducting behavior cloning over intervened rollouts.

**Distillation.** For distillation-based baselines, we optimize the objective in Eq. (9) with the Teacher in Eq. (29) similar to Meng et al. (2023); Ying et al. (2026) on a diverse spectrum of rollouts. Distillation is performed entirely offline without environment interaction.

*Table 7.* Hyper-parameter Settings for PACT. Each base policy utilizes a distinct sampling scheduler.

| Parameter | DP (Chi et al., 2025) | DP3 (Ze et al., 2024) | RDT-1B (Liu et al., 2025) | $\pi_{0.5}$ (Black et al., 2025a) |
|---|---|---|---|---|
| horizon | 3 | 8 | - | - |
| n.obs steps | 3 | 3 | 2 | 1 |
| n.action steps | 6 | 6 | 16 (64) | 16 |
| Inference scheduler | DDPM | DDIM | DPMSolver++ | Flow Matching |
| denoising.pred type | eps | sample | sample | velocity |
| num.inference steps | 100 | 10 | 5 | 5 |
| batch size | 128 | 256 | 32×4 | 32×8 |
| optimizer | AdamW | AdamW | AdamW | AdamW |
| optimizer.betas | [0.95, 0.999] | [0.95, 0.999] | [0.9, 0.999] | [0.9, 0.95] |
| training.use.ema | True | True | True | False |
| obs.use.head camera | True | False | True | True |
| obs.use.left camera | False | False | True | True |
| obs.use.right camera | False | False | True | True |
| obs.use.color point cloud | False | True | False | False |
| action.representation | abs | abs | abs | delta |
| **Base policies training** | | | | |
| optimizer.lr | 1.0e-4 | 1.0e-4 | 1.0e-4 | 2.5e-5 |
| optimizer.lr.scheduler | cosine | cosine | constant | cosine |
| optimizer.lr.warmup steps | 500 | 500 | 500 | 1,000 |
| training.num epochs | 300 | 3000 | - | - |
| training.num iterations | - | - | 10,000 | 45,000 |
| training.gpu | GeForce RTX 4090 | GeForce RTX 4090 | DGX B200 | DGX B200 |
| **Safety alignment distillation** | | | | |
| rollout.num | 288 | 288 | 288 | 288 |
| rollout.use.soft decay | False | True | False | False |
| optimizer.lr | 1.0e-5 | 1.0e-5 | 1.0e-4 | 2.5e-4 |
| optimizer.lr.scheduler | constant | constant | constant | constant |
| training.num epochs | 20 | 20 | 20 | 20 |
| training.num inner epochs | 100 | 20 | 25 | 25 |
| training.gpu | GeForce RTX 5090 | GeForce RTX 5090 | GeForce RTX 5090 | GeForce RTX 5090 |

*Table 8.* Training hyperparameters for all methods in Table 3.

| Method | Data Type | Learning Rate | LR Scheduler | Warmup Steps | Epochs |
|---|---|---|---|---|---|
| *Imitation Learning* | | | | | |
| Guided Rollouts | Guided Rollouts | $1\times10^{-4}$ | Cosine | 500 | 600 |
| PLD | Intervened Rollouts | | | | |
| *Reinforcement Learning* | | | | | |
| iDQL | Self & Guided Rollouts | $1\times10^{-4}$ | Cosine | 500 | 600 |
| *Distillation* | | | | | |
| Expert Rollouts | Expert Rollouts | $1\times10^{-5}$ | Constant | 500 | 600 |
| Self Rollouts | Self Rollouts | | | | |
| Guided Rollouts | Guided Rollouts | | | | |

**Implicit Diffusion Q-learning (iDQL).** We implement iDQL (Hansen-Estruch et al., 2023) on top of the same DP. The training data consists of a mixture of 1,000 *Self Rollouts* collected from the base policy $\varepsilon_\phi$ and 1,000 *Guided Rollouts*, which are used to expand the support of the base policy distribution. For action selection, we follow the official implementation and adopt deterministic sampling, where a batch of candidate actions is generated and the action with the lowest cost is selected for execution with a sampling batch size of 32.

### C.7. Implementation of On-Policy Baselines.

**Overview.** For on-policy alignment baselines, policy updates are performed using data collected online from the current policy, optionally augmented with interventions from an implicit safe teacher in the online variant of PLD. All on-policy baselines are initialized from the same pre-trained DP base policy to ensure a controlled and fair comparison. To align the overall optimization budget with off-policy methods, we fix both the total number of environment interaction steps and the number of policy gradient updates across all baselines. For methods that require a Q-function, we replace it with the negative aggregated cost function as a ground-truth supervision signal, facilitating the training of the baseline while ensuring a fair head-to-head comparison. State–action advantages are estimated using the REINFORCE estimator (Williams, 1992); specifically, the advantage is computed as the negative aggregated safety cost of a given state–action pair, minus the mean safety cost of the training data in the current iteration. This formulation has been shown to improve numerical stability (Ahmadian et al., 2024) without requiring the learning of an explicit Q-function or value model. Unless otherwise specified, the optimizer, network architecture, diffusion horizon, and all other hyperparameters are kept identical to those used in the implementation of PACT on the DP policy (Table 7).

**Training protocol.** For all on-policy baselines, rollouts and updates are interleaved in fixed-length iterations. The total number of environment steps, update epochs per iteration, and overall training steps are matched across methods. Hyperparameters for each on-policy baseline are the same as ones used in the implementations of our method (Table 7) if not otherwise stated.

**Online Rejection Fine-Tuning (Online RFT).** Online RFT is implemented as an IL-style on-policy baseline that repeatedly alternates between policy rollout and supervised fine-tuning. At each iteration, the current policy $\varepsilon_\theta$ is used to collect fresh rollouts. The policy is updated via behavior cloning only on the successful and safe episodes with all the failed episodes dropped out. This procedure can be viewed as on-policy supervised fine-tuning with continuously refreshed data, without explicit value estimation or reward optimization, but requires labor for trajectory annotation.

**Online Probe, Learn, Distill (PLD_online / DAgger).** Online-PLD extends PLD to the on-policy setting and is closely related to Dataset Aggregation (DAgger) (Ross et al., 2011). During each iteration, the base policy $\varepsilon_\phi$ is rolled out in the environment with base policy probing (Xiao et al., 2026), and then corrected by the Implicit Safe Teacher $\varepsilon_\phi^*$. The resulting successful intervened rollouts are aggregated with previously collected data, and the policy is updated via behavior cloning on the aggregated dataset. We adopt the same probing horizon as in the implementation of the original PLD.

**Advantage-Weighted Regression (AWR).** Advantage-Weighted Regression (AWR) (Peng et al., 2019) is an RL method that optimizes a weighted behavior cloning objective using off-policy rollouts and has been widely adopted in the RL of diffusion policies (Zheng et al., 2024; Amin et al., 2025). We extend AWR to an on-policy variant with multiple training iterations to further improve performance. The policy is then updated by regressing toward sampled actions weighted by the exponentiated advantage, following the standard AWR formulation. This procedure enables stable on-policy improvement without explicit policy gradient estimation. Following the hyperparameters of the official implementation, we fix the Lagrange multiplier to $\beta_{\text{awr}} = 0.05$ and apply weight clipping with a threshold of $w_{\max} = 20$ to mitigate exploding weights.

**Q-Score Matching (QSM).** QSM incorporates learned action-value guidance into diffusion policy training (Psenka et al., 2024). The diffusion policy is then optimized via score matching with an additional guidance term proportional to the gradient of the Q-function with respect to the action. This encourages the denoising process to favor high-value actions while remaining close to the behavior induced by the current policy. In our implementation, we set the weight of the Jacobian matrix of Q-function $\alpha_{\text{qsm}} = 6.0$ to match the numerical range of the predicted score to ensure training stability.

**Model-free Reinforcement Learning with Diffusion Policy (DIPO).** We implement DIPO as an online reinforcement learning baseline that performs policy improvement via action-gradient updates rather than conventional policy gradients,

following Algorithm 2 in Yang et al. (2023) with a replaced Q-function. To ensure a fair head-to-head comparison, we use the same action-gradient weighting coefficient as in our method.

**Proximal Policy Optimization (PPO).** We implement a variant of Proximal Policy Optimization (PPO) (Schulman et al., 2017) related to DPPO (Ren et al., 2025), SPO (Xie et al., 2025); we maintain the default hyper-parameters following the recent implementation by (Amin et al., 2025). Specifically, we leave out the term for the autoregressive policy component in the training objective. Log-likelihood estimation mirrors the diffusion likelihood bound by McAllister et al. (2026); Liu et al. (2026a).

## D. Implementation Details for Real World Evaluation

### D.1. Environment and Hardware Setup

All experiments use the Cobot Magic from AgileX Robotics (https://global.agilex.ai/products/cobot-magic) with the configuration of Mobile ALOHA (Fu et al., 2025), featuring two 6-DoF PiPER arms (https://global.agilex.ai/products/piper) as shown in Fig. 9. Scene RGB perception is provided by a head camera and two wrist-mounted cameras of the Intel D435i camera. The inferences of real-world experiments are performed on a single RTX 4090 GPU. The technical specifications of the selected model are listed in Table 9. To access privileged state information for cost function computation, we use an Intel L515 as an exterior camera for high-fidelity point clouds. It is worth noting that we used the "mobile" ALOHA only to facilitate transportation and do not use its autonomous mobility feature during any training or inference stage. Our tasks are still static manipulation tasks.

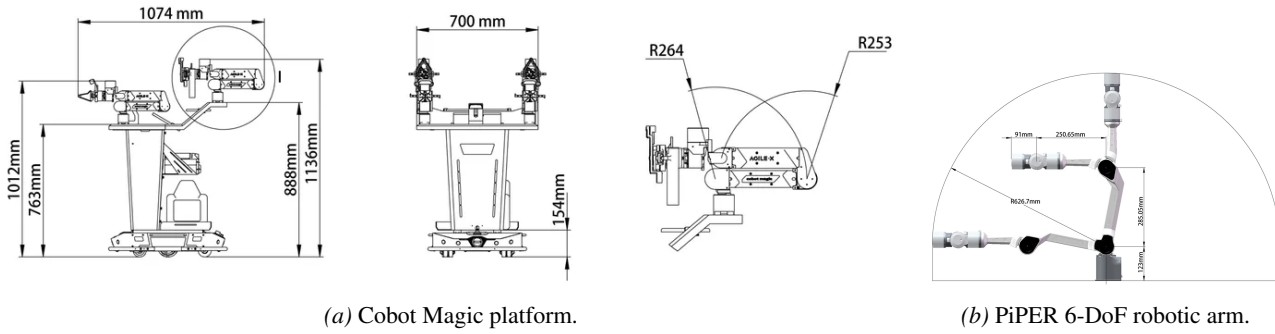

*(a)* Cobot Magic platform.  *(b)* PiPER 6-DoF robotic arm.

*Figure 9.* Cobot Magic mobile manipulator with the Mobile ALOHA configuration and dual PiPER arms.

*Table 9.* **Technical specifications of AgileX Cobot Magic.**

| Parameter | Value |
|---|---|
| DoF | $6 \times 2 = 12$ |
| Payload | 1.5 kg |
| Arm weight | 4.2 kg |
| Arm repeatability | $\pm 0.1$ mm |
| Arm reach | 626 mm |
| Material | Aluminum alloy body & Polymer shell |
| Arm working radius | 626.7 mm |
| Gripper range | 0-100 mm |

### D.2. Task Description

As illustrated in Fig. 10, we evaluate PACT on four real-world manipulation tasks, namely pour water, nail insertion, transfer egg, and GPU assembly. These tasks are deliberately chosen for their safety-critical attributes. Each task involves physical interactions where failures could lead to material damage, equipment harm, or hazardous situations, thereby demanding high reliability and precise control from the robotic system. For instance, the manipulation of liquids and fragile objects—such as

water and eggs—requires precise handling. Improper grasping or misalignment can lead to eggshell rupture and content leakage. In practice, we do not perform real liquid pouring, but instead employ a capped-bottle setup as a proxy pouring task. This design substantially simplifies experimental reset and improves evaluation consistency, since real liquid pouring would introduce significant overhead and uncontrolled variability across trials, including bottle refilling, spill handling, and workspace cleaning. Accordingly, the constraint in this task is defined by whether the bottle is properly aligned above the cup during the pouring phase, which serves as a geometric proxy for preventing potential spillage. Among all evaluated tasks, GPU Assembly (RTX 2080Ti) is particularly safety-critical which involves handling delicate, expensive computer components. Misplacement or applying sideways force during placement could damage its electric components, leading to significant financial cost and functional failure. By focusing on such tasks, our evaluation directly assesses a robotic policy's ability to perform reliable, precise, and physically-aware manipulation under constraints where errors have clear and undesirable consequences.

The physical constraints activated per task are illustrated in Table 10.

*Table 10.* Activated physical constraints for each task in real world evaluation.

| Task | Gripper Poking Cost | Behavior Alignment Cost | Gripper Rotation Cost |
|---|---|---|---|
| Transfer Egg | ✓ | | |
| Nail Insertion | | ✓ | |
| Pour Water | ✓ | ✓ | |
| GPU Assembly | ✓ | ✓ | ✓ |

### D.3. Evaluation Rubric

We follow prior real-world evaluations in (Black et al., 2025b;a) and use the average percentage of achieved points as our real-world metric for task-execution. For each real-world manipulation task, we evaluate PACT over 25 trials with randomized initial object placements to assess robustness. The task-specific scoring criteria are as follows:

- **Pour Water:** The task starts with a bottle and a cup on the table. The goal is to pour water from the bottle into the cup.
  +1 Grasping the bottle.
  +1 Lifting the bottle without dropping it.
  +1 Tilting the bottle by more than $60°$ to initiate pouring.
  +1 Aligning the bottle mouth with the cup rim.
  *Maximum score: 4 points.*

- **Nail Insertion:** The task starts with a nail placed on a soft putty base on the table. The robot must use a gripper finger to press the nail downward into the putty.
  +1 Aligning the gripper finger with the nail head.
  +1 Pressing the nail vertically into the putty.
  *Maximum score: 2 points.*

- **Transfer egg:** The task starts with eggs on a plate and an egg tray on the table. The goal is to transfer an egg from the plate to the tray.
  +1 Grasping an egg from the plate.
  +1 Lifting and moving the egg to the tray without dropping it.
  +1 Releasing the egg into the tray.
  *Maximum score: 3 points.*

- **GPU assembly:** The task starts with a GPU and a heat sink on the table. The goal is to pick up the heat sink and place it onto the GPU.
  +1 Grasping the heat sink.
  +1 Lifting and moving the heat sink above the GPU without dropping it.
  +1 Placing the heat sink at the aligned position on the GPU.
  +1 Releasing the heat sink.
  *Maximum score: 4 points.*

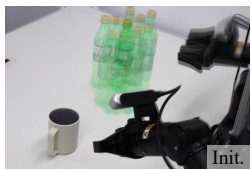 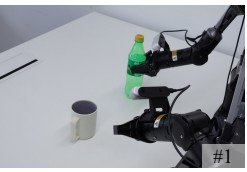 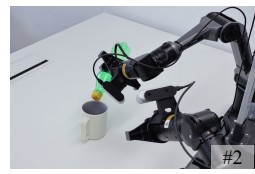 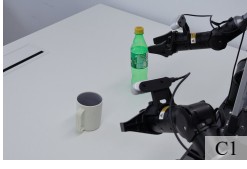 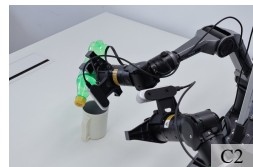

**Pour Water:** The bottle are randomized within region of 25 cm × 25 cm. The robot needs to Pick Up Bottle (#1), left it and Pour Water Right into Cup (#2). Two physical constraints are enforced: (C1) **Gripper Poking**, which prevents unintended contact that could _destabilize or knock over the bottle,_ and (C2) **Behavior Alignment**, which requires the bottle to be held directly above the cup to avoid _spillage_ during the pouring phase.

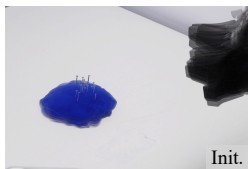 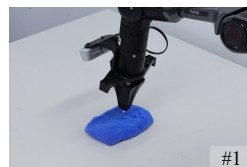 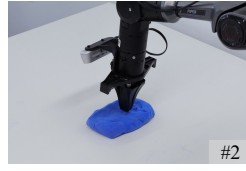 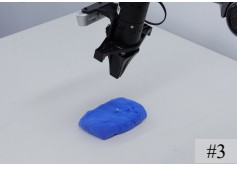 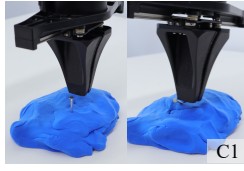

**Nail Insertion:** The nail is placed at a randomized position on a soft putty base (15 cm × 15 cm) within the workspace. The robot is required to aim the nail toward the putty (#1), drive it into the putty through controlled downward motions (#2), and ensure successful insertion (#3). One physical constraint is enforced: (C1) **Behavior Alignment**, which ensures that the gripper is vertically aligned with the putty surface and positioned directly above the nail, enabling stable insertion while avoiding _collisions with irrelevant surfaces_ or _unintended tilting of nail_.

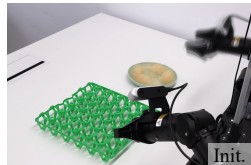 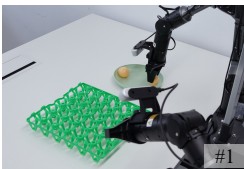 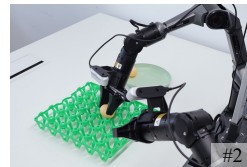 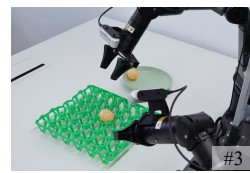 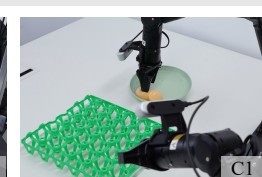

**Transfer Egg:** The eggs are randomized within the plate (circular region of radius 8 cm). The robot needs to delicately pick up one egg from the plate (#1) and transfer it into the egg tray (#2–#3) without causing damage. One physical constraint is enforced: (C1) **Gripper Poking**, which limits excessive contact forces during grasping to prevent _cracking the egg_.

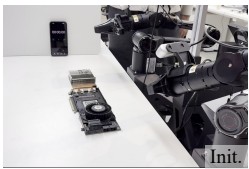 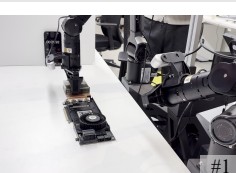 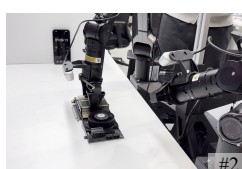 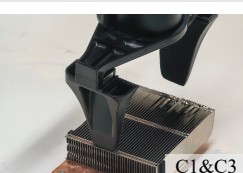 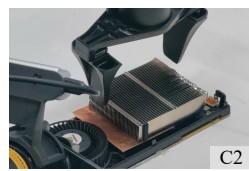

**GPU Assembly:** The GPU and heat sink are placed within the assembly area, with the heat sink position randomized by up to 5 cm. The robot is required to pick up the heat sink (#1) and align it while inserting it onto the GPU (#2) with millimeter-level precision. Two physical constraints are enforced: (C1) **Gripper Poking**, which regulates the approach direction to avoid unintended contact that could _damage the heat sink,_ and (C2) **Behavior Alignment**, which ensures _precise pose alignment between the heat sink and the GPU_ during insertion. (C3) **Gripper Rotation**, which controls controls the orientation about the approach axis, ensuring that the gripper fingers remain parallel to the block surface at closure to prevents undesirable torque generation during contact which may induce _rotational slippage of the heat sink._

_Figure 10._ **Task definitions and visualizations.** For 4 safety-critical tasks, we describe their randomization, definitions of each sub-task, and corresponding physical constraints.

We use Safe Rate to quantify safety during real-world evaluation. It is worth noting that success and safety are correlated to some extent, yet remain conceptually distinct: success measures task completion, while safety avoids predefined unsafe interactions. For instance, a rollout may complete a task with unsafe contact (e.g., poking an egg before successfully grasping it), or remain safe but fail (e.g., not reaching any object). Following Appendix C.4, we define three generic hazardous events including poking, falling, and toppling that capture the most common unsafe outcomes in manipulation, and we manually annotate their occurrences during real-world rollouts. A trial is deemed safe if none of these events are triggered throughout the episode. Accordingly, human annotators assess both safety and task success for each rollout based on the pre-defined rubrics, following common evaluation protocols adopted in concurrent real-world manipulation studies (Yakefu et al., 2025), thereby ensuring consistent and objective evaluation. We then compute the safe rate as the fraction of safe trials averaged over the evaluation episodes for each task.

### D.4. Implementation of PACT

**Model.**   For real world experiments, we adopt the RDT2 (Liu et al., 2026b) with its official implementation at `https://github.com/thu-ml/RDT2/tree/main` and replace the Qwen2.5-VL (Bai et al., 2025) with a DINOv3 vision encoder (Siméoni et al., 2025) to improve the performance on a single real world task (Bi et al., 2026). Following the Transformer Backbone, all adaptors for multi-modal inputs use Sigmoid-Weighted Linear Unit (SiLU) activation.

*Table 11.* Model architecture and configuration summary.

| Parameter | Value |
|---|---|
| Image history size | 1 |
| Action chunk size | 24 |
| Action dimension | $7\times2=14$ |
| State dimension | $7\times2=14$ |
| Action representation | absolute joint angle |
| System frequency | 30 Hz |
| *Input Adaptors* | |
| Image adaptor | MLP (2 layers, SiLU) |
| Action adaptor | MLP (3 layers, SiLU) |
| State adaptor | MLP (3 layers, SiLU) |
| *RDT Backbone* | |
| Hidden size | 1024 |
| Layers | 14 |
| Attention heads | 16 |
| KV heads (GQA) | 8 |
| Register tokens | 4 |
| FFN multiple | 256 |
| Normalization $\epsilon$ | $1 \times 10^{-5}$ |
| *Diffusion Modeling* | |
| Diffusion Parameterization | Flow matching |
| Inference steps | 5 |
| **Total parameters** | **498.05M** |

**Training Configuration of Base Policy.**   For each task, we collect 200 demonstrations with tele-operation to train a corresponding model from scratch. Similar to the implementation of RDT-1B, the vision encoder is frozen during training. The hyper-parameters for training are listed in Table 12.

**Implementation of Safety Alignment.**   Following the implementation of aligning RDT-1B in simulation, we use LoRA to fine-tune the Transformer backbone with the same configuration as RDT-1B for simulation, and other hyperparameters are elaborated in Table 12 as well.

*Table 12.* Hyper-parameter settings for real world evaluation.

| Parameter | Value |
|---|---|
| Optimizer | AdamW |
| Optimizer betas | [0.9, 0.9999] |
| **Base policy training** | |
| batch size | $64 \times 8$ |
| Learning rate | $1.0 \times 10^{-4}$ |
| Learning-rate scheduler | Constant |
| Warmup steps | 500 |
| Training iterations | 50,000 |
| Training GPU | DGX B200 |
| **Safety alignment distillation** | |
| batch size | $32 \times 8$ |
| Number of rollouts | 40 |
| Learning rate | $1.0 \times 10^{-4}$ |
| Learning-rate scheduler | Constant |
| Training epochs | 8 |
| Inner epochs | 25 |
| Training GPU | DGX B200 |

# E. Additional Results

### E.1. Robustness Against Compromised Privileged State Information

Since PACT relies on privileged state information to compute safety costs and their gradients during distillation, it is necessary to assess how sensitive alignment is to corruption in these inputs, as would occur with imperfect upstream perception. To this end, we evaluate the robustness of PACT to errors in the revealed privileged state information used to construct safety costs and their gradients. Specifically, we inject zero-mean Gaussian noise into privileged states during post-training, where the noise scale is defined as the standard deviation of the added noise normalized by the norm of the original privileged states. As shown in Fig. 11, we find that PACT remains effective under moderate corruption, sustaining strong success and safety performance up to a noise magnitude of 12% of the original state norm (12% corresponds to approximately $\pm 2$ cm deviations), which suggests that PACT does not require perfect perception to be useful.

### E.2. Qualitative Results for Simulation Evaluation

We further present qualitative results for three simulated tasks in Fig. 12. The base DP policy (top) exhibits typical safety violations, including gripper poking, behavior misalignment, and incorrect gripper rotation, which often lead to failures. After post-training with PACT (bottom), these failure modes are largely corrected, yielding safer interactions and higher task completion as demonstrated in Table 1.

### E.3. Comparison to Inference-Time Baselines.

To enable more rigorous evaluations, we additionally compare PACT with JM2D (Jung et al., 2025), an inference-time guardrail method designed to improve task performance while maintaining constraint satisfaction through mutual compatibility modeling. As shown in Table 13, PACT consistently achieves higher Success and Safe Rates across all tasks, suggesting that directly distilling safety-aware behaviors into the policy yields stronger and more stable safety-performance trade-offs than relying solely on test-time action filtering or corrections.

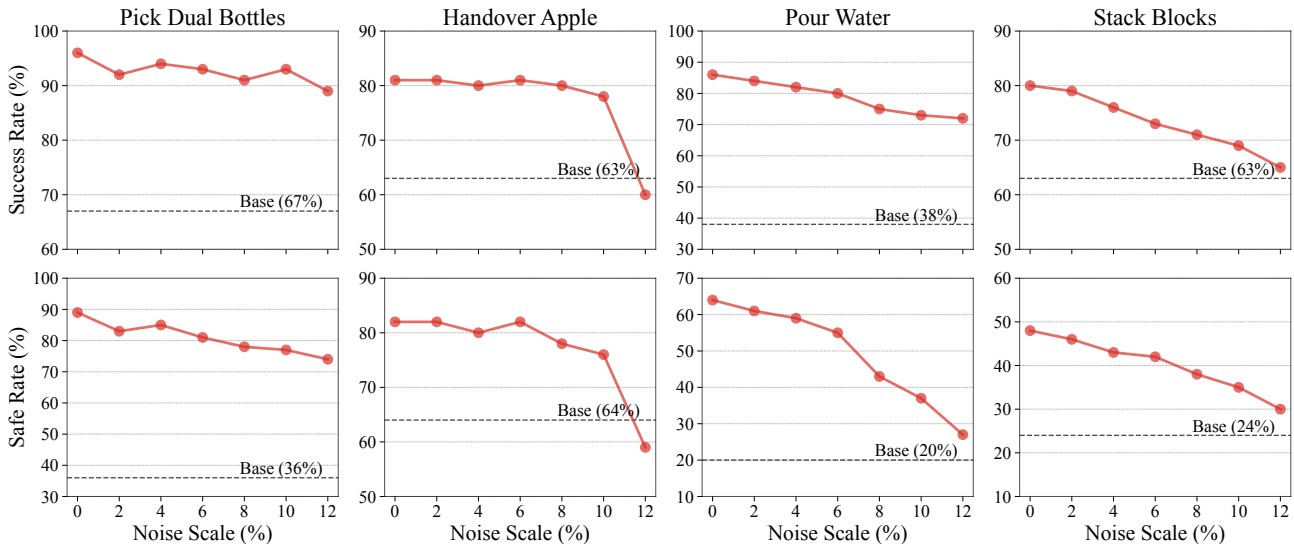

*Figure 11.* **Robustness to noisy privileged state information.** Success rates (top) and safe rates (bottom) under increasing Gaussian noise injected into the privileged state information used for safety supervision. The noise scale is defined as the standard deviation of the injected noise normalized by the norm of the original privileged states. The horizontal dashed lines denote the corresponding pretrained base policy performance without post-training alignment.

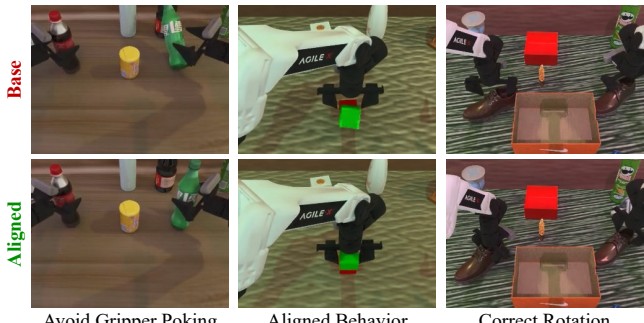

|  |  |  |
|---|---|---|
| Avoid Gripper Poking | Aligned Behavior | Correct Rotation |

*Figure 12.* **Qualitative results in simulation.** Base policy (top) vs. Aligned Policy (bottom) on *Pick Dual Bottles*, *Stack Blocks*, and *Place Dual Shoes* (left-to-right). PACT reduces unsafe behaviors including poking, misalignment in both position and gripper pose.

### E.4. Effect of Base Policy Quality on Post-Training Gains

The magnitude of improvement depends on the competence of the initial policy. We observe smaller absolute gains for weaker base policies (e.g., DP3), whose failures are often dominated by missing core skills, leaving limited scope for safety-aware refinement. In contrast, stronger policies (e.g., RDT-1B) benefit more consistently, suggesting that PACT primarily serves as a post-training safety and reliability enhancer that is most effective when the base policy already possesses reasonable task understanding, and the remaining errors stem from unsafe dynamics or marginal failure cases. Improvements are also more pronounced on difficult, precision-critical tasks (e.g., Pour Water and Stack Blocks), where minor control errors can readily trigger safety violations and thus strongly affect both Succ. and Safe. Conversely, for relatively simple or near-saturated tasks (e.g., Handover Apple), baseline performance is already high and PACT yields only marginal gains. Notably, in tasks with extremely tight safety margins (e.g., stacking), Safe may remain a bottleneck even as Succ. improves, reflecting the strictness of the safety criteria rather than incomplete task execution.

### E.5. Results of Collapsed On-Policy Baselines

We also attempted to apply representative on-policy baselines, AWR (Peng et al., 2019), RFT (Gilks & Wild, 1992), and QSM (Psenka et al., 2024) to our setting. However, all three methods exhibited pronounced training instability despite extra efforts for training stability introduced, including soft updates (Haarnoja et al., 2018) and a KL regularizer (Schulman

*Table 13.* **Performance comparison with Inference-time baselines.** Both the Success Rate (Succ.) and Safe Rate (Safe) are reported.

| Method | Pick Dual Bottles | | Handover Apple | | Pour Water | | Stack Blocks | |
| --- | --- | --- | --- | --- | --- | --- | --- | --- |
| | Succ. | Safe | Succ. | Safe | Succ. | Safe | Succ. | Safe |
| *Imitation Learning* | | | | | | | | |
| Base | 67% | 36% | 63% | 64% | 38% | 20% | 63% | 24% |
| *Inference-time Guardrails* | | | | | | | | |
| JM2D | 87% | 66% | 65% | 68% | 75% | 40% | 76% | 27% |
| *Distillation* | | | | | | | | |
| Ours | **96%** | **89%** | **81%** | **82%** | **86%** | **64%** | **80%** | **48%** |

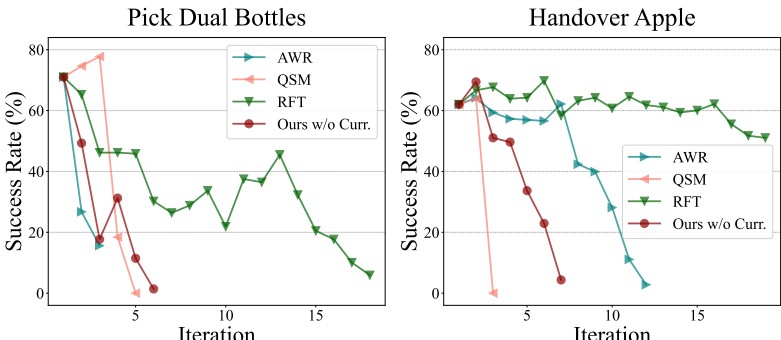

*Figure 13.* **Success rate (%) of collapsed on-policy baselines.** Success rate versus training iteration for three on-policy methods (AWR, QSM, and RFT) and our ablation *Ours w/o Curr.* on *Pick Dual Bottles* (left) and *Handover Apple* (right). Although some methods achieve moderate success in early iterations, performance is highly unstable: AWR and QSM rapidly collapse to near-zero success, and RFT shows substantial degradation over training (especially on *Pick Dual Bottles*), preventing a meaningful comparison in the main results. Meanwhile, *Ours w/o Curr.* also degrades and eventually gets near zero success.

et al., 2017). As shown in Fig. 13, they may reach moderate success rates in the initial iterations, yet performance quickly degrades with continued training and ultimately collapses to low (and in some cases near-zero) success. Consequently, we omit these on-policy baselines from the main comparisons since their collapse precludes a meaningful evaluation.

We further analyze the collapse of these on-policy baselines and find that it is consistent with a self-reinforcing distribution-drift failure mode. AWR and RFT are weighted imitation learning methods; RFT can be viewed as a binarized variant of AWR that retains samples above an advantage threshold. In our safety-alignment regime, advantages estimated from trajectory costs are noisy, so exponentiation (AWR) or hard selection (RFT) induces severe weight degeneracy and sharply reduces the effective sample size. As a result, updates concentrate on a small, non-representative subset of rollouts, accelerating support shrinkage and triggering catastrophic forgetting of useful skills. Through closed-loop rollouts, this policy shift compounds over iterations and quickly drives the state distribution into safe-but-unproductive regions, where task-utility supervision is unavailable to recover competence, rendering the degradation effectively irreversible. By comparison, $\text{PLD}_{\text{online}}$ mitigates collapse by introducing intervention of the implicit safe teacher at the end of each rollout, and provides external safe and useful behaviors as a supervision signal to preserve task-relevant behavior.

Meanwhile, QSM exhibits a related instability: its learned $Q$-function is queried on noisy diffusion-time inputs, yielding high-variance gradients and numerically brittle supervision. Moreover, QSM can be interpreted as a special case of direct distillation with a normalized teacher score. Consider the normalized distillation objective

$$\mathcal{J}(\theta) = \mathbb{E}_{t,\epsilon,(s,a)\sim d^{\pi_\theta}} \left\| \epsilon_\theta(a_t, s, t) - \left( \epsilon_\phi(a_t, s, t) - \sum_{k=1}^{m} \lambda_k \nabla_{a_t} c_{k,t}(s, a_t)/\sigma_t \right) / \sum_{k=1}^{m} \lambda_k \right\|^2 \tag{30}$$

When the multipliers are uniformly large, $\lambda_k \to \infty$ with a comparable scale, the base-score term vanishes after normalization

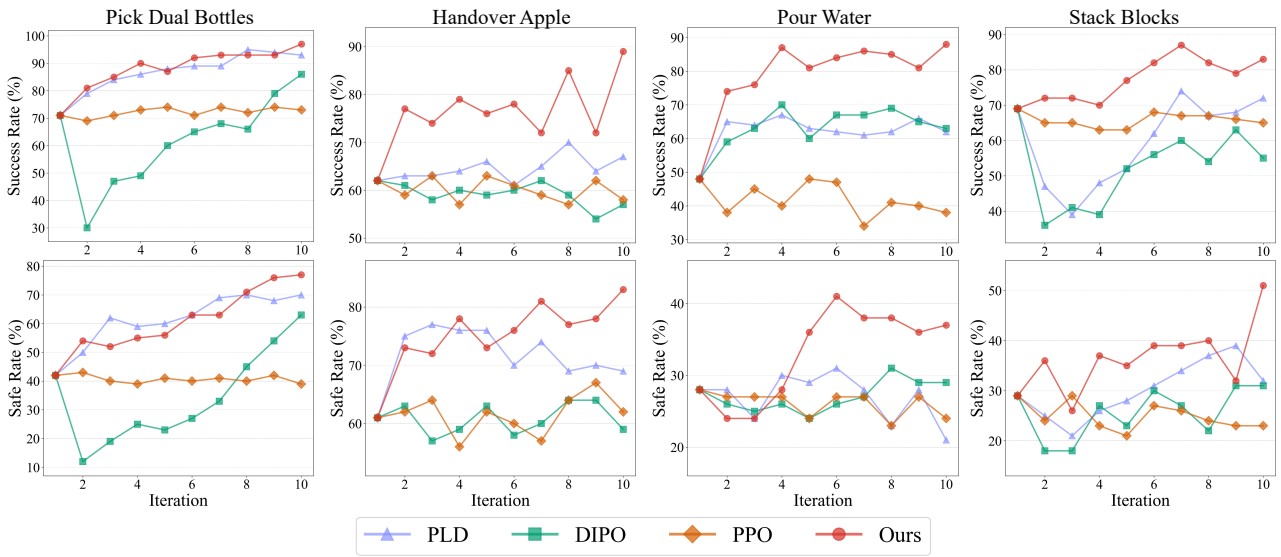

*Figure 14.* **Training efficiency comparison with on-policy baselines.** Success Rate (top) and Safe Rate (bottom) over training iterations on four tasks: *Pick Dual Bottles*, *Handover Apple*, *Pour Water*, and *Stack Blocks*. Compared with PLD, DIPO, and PPO, our method reaches higher performance in fewer iterations and yields more stable training curves, achieving the best final Success/Safe Rates across tasks.

and the target reduces to a (weighted) average constraint-gradient direction, giving

$$
\begin{aligned}
\mathcal{J}(\theta) &= \mathbb{E}_{t,\boldsymbol{\epsilon},(\boldsymbol{s},\boldsymbol{a})\sim d^{\pi_\theta}} \left\| \boldsymbol{\epsilon}_\theta(\boldsymbol{a}_t,\boldsymbol{s},t) - \left( \boldsymbol{\epsilon}_\phi(\boldsymbol{a}_t,\boldsymbol{s},t) - \sum_{k=1}^{m} \lambda_k \nabla_{\boldsymbol{a}_t} c_{k,t}(\boldsymbol{s},\boldsymbol{a}_t)/\sigma_t \right) / \sum_{k=1}^{m} \lambda_k \right\|^2 \\
&= \mathbb{E}_{t,\boldsymbol{\epsilon},(\boldsymbol{s},\boldsymbol{a})\sim d^{\pi_\theta}} \left\| \boldsymbol{\epsilon}_\theta(\boldsymbol{a}_t,\boldsymbol{s},t) + \frac{1}{m}\sum_{k=1}^{m} \nabla_{\boldsymbol{a}_t} c_{k,t}(\boldsymbol{s},\boldsymbol{a}_t)/\sigma_t \right\|^2 \\
&= \mathcal{J}_{\text{QSM}}(\theta)
\end{aligned}
\tag{31}
$$

Consequently, excessively large multipliers correspond to aggressive constraint enforcement and can induce collapse, consistent with our ablations in Sec. 4. In addition, mismatched numerical scales between predicted scores and cost gradients can further destabilize optimization, exacerbating the sensitivity of the QSM.

### E.6. Detailed Training Efficiency Comparison with On-Policy Baselines

We compare the training dynamics of PACT with three on-policy baselines in Fig. 14 on four tasks: *Pick Dual Bottles*, *Handover Apple*, *Pour Water*, and *Stack Blocks*. Specifically, we track both task success rate and safety rate throughout training. PACT consistently exhibits superior training efficiency, achieving higher Success and Safe Rates earlier and attaining the best final performance across all four tasks.

### E.7. Inference Time Analysis

We additionally provide the inference time analysis of the evaluated policies. One key advantage of our approach is that safety alignment is performed entirely during post-training rather than at deployment time. Consequently, *the aligned policy preserves exactly the same inference procedure as the original base policy*, without requiring any additional test-time optimization, action filtering, or guidance injection during rollout. Therefore, our method introduces essentially no additional inference overhead compared with the corresponding pretrained policy. Table 14 reports the average inference time for generating one action chunk for each evaluated policy. The measurements are conducted on the same hardware platform used in our real-world experiments.

These results further demonstrate that our method improves safety alignment without sacrificing deployment efficiency, making it suitable for real-world robotic manipulation settings where low-latency inference is important.

*Table 14.* Average inference time (ms) per action chunk.

| Model | DP | DP3 | RDT | $\pi_{0.5}$ |
|---|---|---|---|---|
| Inference Time | 455.53 | 49.84 | 179.43 | 91.83 |

### E.8. More Ablation Studies

**Curriculum distillation.** We initially investigate the effect of curriculum safety constraints during training, which explicitly define a smooth transition between the base and the aligned policy by introducing a guidance schedule. We compare direct constraint distillation (Eq. (9)) against progressively tightening the constraint multipliers using a simple linear schedule $\eta_i = i/N$ as described in Eq. (11). We find that direct distillation frequently induces abrupt distributional shifts, leading to the collapse of the action support and noticeable performance drops consistent with catastrophic forgetting (*Ours w/o Curr.* in Fig. 13). In contrast, curriculum-based enforcement yields smoother optimization dynamics and maintains continuous improvement.

**Impact of Lagrange multipliers $\lambda$.** The Lagrange multiplier $\lambda$ controls the strength of constraint guidance during alignment. As shown in Fig. 6 with grid search, the guidance strength exhibits a clear trade-off between constraint enforcement and policy stability. When the multiplier is excessively large (e.g., $\times 5$ or $\times 10$), the policy collapses across most tasks, failing to make meaningful progress, which corresponds to direct distillation that enforces constraints and leads to irreversible OOD collapse. In contrast, extremely small multipliers ($\times 0.1$) lead to under-enforced constraints; the alignment process becomes inefficient and yields limited safety improvements despite being stable. The optimal performance consistently emerges near the turning point of this trade-off, which enables simple hyperparameter selection. These results validate the necessity of curriculum distillation and highlight the robustness of PACT within a reasonable and practical multiplier range.

**Efficient distillation with few diffusion steps $t_c$.** As shown in Fig. 7, increasing the number of guided denoising steps does not monotonically improve performance. Our experiments demonstrate that injecting constraint guidance for only a few denoising steps at the late stage (e.g., $t_c = 0.03$) is sufficient to achieve strong alignment, whereas more aggressive distillation ($t_c \geq 0.1$) often degrades both task success and safety performance.

This behavior can be attributed to the noise structure of diffusion models. Early denoising steps correspond to high-noise regimes, particularly when $t_c \rightarrow 1$, where sampled actions are dominated by near-Gaussian randomness. Applying safety costs at this stage yields gradients that are weakly correlated with the final action realization and may significantly distort the original diffusion sampling distribution, thereby impairing stable and meaningful policy optimization. These findings empirically justify our efficient design choice of late-stage, few-step constraint distillation, which aligns well with prior observations in guided diffusion while preserving the integrity of the pretrained policy distribution.

**Sensitivity to number of rollouts.** Our main simulation comparisons (Sec. 4.2) use a relatively large rollout budget to place all baselines, particularly RL-based methods that are typically interaction-intensive, in a sufficiently informative data regime for fair head-to-head evaluation, rather than to suggest a minimum requirement for PACT. To assess practical scalability to low-data regime, we additionally rerun post-training under reduced rollout budget. As shown in Fig. 15, PACT degrades gracefully as the budget decreases and remains effective in low-data settings across tasks, whereas RL-based baselines exhibit substantially larger performance drops and higher variance. This behavior is consistent with the design of PACT: constraint distillation leverages differentiable cost gradients to provide dense, direct supervision, which is typically more sample-efficient than sparse, trajectory-level reinforcement signals.

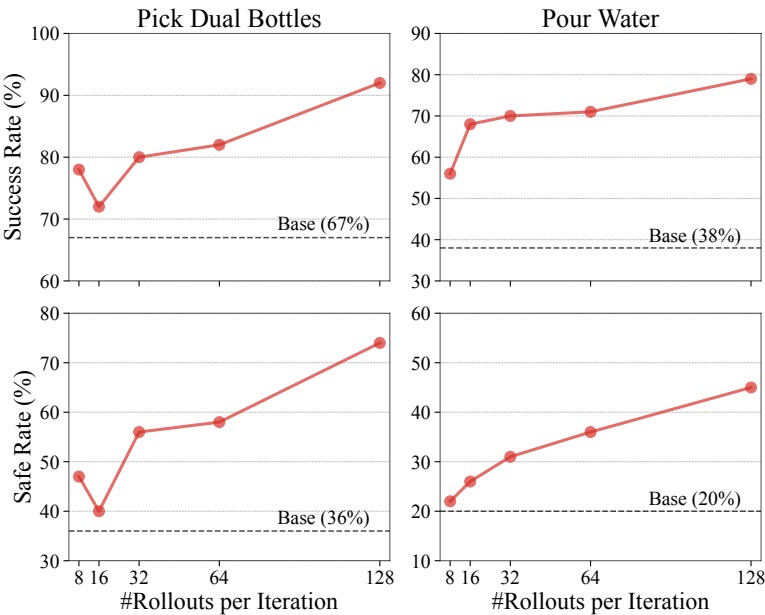

*Figure 15.* **Low-data post-training with varying rollout budgets.** Success rates (top) and safe rates (bottom) under different rollout collection budgets, where all settings use 10 post-training iterations. The horizontal dashed lines indicate the corresponding pretrained base policy performance.

