# OpenReview forum: "PACT: Self-Evolving Physical Safety Alignment for Diffusion Policies in Embodied Manipulation"
_ICML.cc/2026/Conference — ICML 2026 spotlight_

### Official Review · Reviewer_8sC6 · 2026-02-24

**Soundness:** 3
**Presentation:** 3
**Significance:** 3
**Originality:** 3
**Overall Recommendation:** 5
**Confidence:** 3

**Summary:**

This work introduces PACT, a post-training framework for aligning Diffusion policies to safety constraints.

The proposed method is evaluted in simulation (RoboTwin) and on several real robot experiments against a variety of Imitation Learning and Reinforcement learning baselines, both on- and off-policy.

**Compliance With Llm Reviewing Policy:**

Affirmed.

**Final Justification:**

The rebuttal answered my open questions. I believe this work addresses the important problem of safety alignment, and I recommend to accept this work.

**Key Questions For Authors:**

As stated in the weaknesses section, I would like clarifications on:
1. How many rollouts / iterations of PACT were required for post training on the real world experiments?
2. How does PACT scale within a low-data regime? 1000 rollouts in simulation seem very high. I would welcome an analysis similar to Figure 4, but with a rollout budget << 1000.

**Limitations:**

Yes. I especially appreciate and want to highlight the authors' foresight to advocate against treating this and related alignment methods as safety guarantees in critical applications.

**Strengths And Weaknesses:**

## Strengths
- well written and good presentation
- solid theoretical foundation
- thorough experimental evaluation with a wide range of relevant baselines and tasks, both in sim and real robot.
- detailed appendix aids in potentially reproducing the work

## Weaknesses
- It is unclear to me, how much data / how many rollouts the method requires.
Figure 4 details a higher efficiency with regard to iterations. However, 1000 rollouts per task in simulation seem very high. How does the method scale in low-data regime, for example 10 iterations with 5 rollouts each --> 50 rollouts total.
- How many post training iterations and rollouts were required in the real robot experiments?

---

> ### Author Rebuttal · Authors · 2026-03-31
>
> -  **W1 & Q2 Rollout amount and scalability to low-data regime**
>
> 	We thank the reviewer for the valuable suggestion about the practical applicability. We agree that 1000 simulation rollouts may appear high if interpreted as a minimum requirement. However, the purpose of Fig. 4 is to compare all baselines under a sufficiently rich data regime so that RL-based baselines, which typically require more samples, can better realize their potential. Thus, the large rollout budget is intended for a fair comparison across methods, rather than to suggest that such a budget is necessary for our approach.  More broadly, this scale is not unusual relative to recent real-world robot post-training systems, which often rely on hundreds of episodes. For example, RL100 reports ~1k episodes [1], while $\pi_{0.6}^*$ uses ~500 demonstrations together with human correction data [2]. To further examine the low-data regime more directly, we additionally evaluate reduced rollout budgets, and observe that **PACT can maintain a sufficient safety and performance gain even under a harsh low-data regime (10 iterations with 8 rollouts each)**.
>
>     _Table: Low-data post-training results with varying rollout budgets, where each setting uses 10 iterations and the column header denotes the number of rollouts collected per iteration_
> 	|       Task / #Rollouts       |  Base   |    8    |   16    |   32    |   64    |   128   |
> 	| :----: | :-----: | :-----: | :-----: | :-----: | :-----: | :-----: |
> 	| Pick dual bottles | 67 / 36 | **78 / 47** | 72 / 40 | 80 / 56 | 82 / 58 | 92 / 74 |
>     | Pour water | 38 / 20 | **56 / 22** | 68 / 26 | 70 / 31 | 71 / 36 | 79 / 45 |
>
> 	We attribute this advantage primarily to our objective distillation formulation, which leverages cost gradients to provide more direct and stable supervision than RL-based methods, thereby achieving better training efficiency under limited data. Moreover, the results and analyses on PACT's adaptability to low-data regime will be updated to the revised version of our paper.
>
> - **W2 & Q1 Rollout amount and iterations for real-world**
>
> 	Thank you for pointing this out. Concretely, PACT uses 8 post-training iterations, with 40 rollouts collected per iteration, resulting in 320 rollouts in total for each real-world task. This rollout budget is lower than that of concurrent real-world RL methods, which we attribute to the higher sample efficiency of our distillation objective. More detailed statistics on the real-world post-training budget are provided in Appx. D.4 and Table 12. We will make this point more explicit in the main text as well.
>
> **References:**
>
> [1] Lei, Kun, et al. "Rl-100: Performant robotic manipulation with real-world reinforcement learning." arXiv preprint arXiv:2510.14830 (2025).
>
> [2] Intelligence, Physical, et al. "$\pi^{*} _ {0.6}$: a VLA That Learns From Experience." arXiv preprint arXiv:2511.14759(2025).

---

> > ### Author Rebuttal · Reviewer_8sC6 · 2026-04-01
> >
> > I sincerely thank the authors for their response and clarifications.
> > My questions have been fully resolved and I will raise my score.

---

> > > ### Author Response · Authors · 2026-04-01
> > >
> > > Thank you very much for your positive feedback and for taking the time to review our rebuttal. We are glad that our clarifications addressed your concerns, and we sincerely appreciate your thoughtful comments and support.

---

### Official Review · Reviewer_sz4p · 2026-03-13

**Soundness:** 3
**Presentation:** 3
**Significance:** 3
**Originality:** 3
**Overall Recommendation:** 4
**Confidence:** 4

**Summary:**

The paper proposes Physical safety Alignment for Constrained Trajectories (PACT), a self-evolving post-training framework designed to align continuous diffusion-based manipulation policies with strict physical safety constraints. Instead of requiring human demonstrations or external reward engineering, PACT operates on self-generated trajectories, projecting pretrained diffusion policies onto CMDP feasible regions. The method utilizes an "Implicit Safety Teacher" to distill constraint gradients into the diffusion model via a reverse-KL objective, providing dense supervision across timesteps. To prevent catastrophic forgetting and Irreversible Out-of-Distribution (OOD) Collapse, PACT introduces a curriculum distillation schedule that progressively tightens constraints, maintaining a theoretically bounded policy shift. The authors validate their approach on both simulated and real-world bimanual manipulation benchmarks, demonstrating a 31.0% reduction in safety violations and a 30.7% improvement in task success rates compared to baseline methods.

**Compliance With Llm Reviewing Policy:**

Affirmed.

**Final Justification:**

Based on the rebuttal and subsequent discussion, I have decided to maintain my initial score

**Key Questions For Authors:**

In Section 3.3, you inject constraints only in the final few denoising steps. If a safety violation requires a macroscopic trajectory correction rather than a late-stage micro-adjustment, does this late-stage injection still reliably steer the policy away from the hazard?

How sensitive is the PACT framework to noise or failure in the 3D perception pipelines used to gather privileged state information during the self-rollout phase?

The curriculum effectively prevents "Irreversible OOD Collapse". However, is there a risk that a slowly tightening curriculum might still get trapped in a local optimum where the policy is safe but fails the task, simply because the safe region is disconnected from the task-success region in the action space?

**Limitations:**

yes

**Strengths And Weaknesses:**

Strengths:

The authors provide formal proofs for the optimality of the PACT distillation. Furthermore, they mathematically guarantee monotonic safety improvement and controlled policy shifts under their curriculum formulation. To address the intractable nature of exact gradient computation for intermediate costs, the authors introduce a training-free approximation using a Taylor expansion around t=0 to evaluate gradients at the posterior mean of the clean action. This serves as a clever and mathematically justified workaround.

Weaknesses:

The methodology's reliance on privileged state information during the data collection and alignment phase heavily depends on the robustness of an external 3D perception pipeline in real-world scenarios. Consequently, if these perception modules fail, misidentify keypoints, or hallucinate during the self-rollout phase, the resulting safety gradients injected into the diffusion process will be fundamentally flawed.

---

> ### Author Rebuttal · Authors · 2026-03-31
>
> - **W1 & Q2 Robustness against compromised state information**
>
> 	We thank the reviewer for pointing this out. To this end, we evaluate robustness to noise in the revealed privileged information, and observe that the method **remains effective under moderate corruption, including noise with magnitude up to 12% of the original data norm** (12% corresponds to an error of ±2cm). This indicates that our method does not require perfect perception to be useful. We will update the evaluation results and analyses into the revised version of our paper.
>
>     *Table: Success/safety rates under Gaussian noise injected into privileged state information, where the noise scale is defined as the standard deviation of the added noise normalized by the norm of the original privileged states.*
> 	| Task \ Noise Scale | Base | PACT | 2% | 4% | 6% | 8% | 10% | 12% |
> 	|---|:---:|:---:|:---:|:---:|:---:|:---:|:---:|:---:|
> 	|  pick dual bottles  |  67/36  |  96/89  |  92/83  |  94/85  |  93/81  |  91/78  |  93/77  |  89/74  |
> 	|  handover apple  |  63/64  |  81/82  |  81/82  |  80/80  |  81/82  |  80/78  |  78/76  |  60/59  |
> 	|  stack blocks  |  63/24  |  80/48  |  79/46  |  76/43  |  73/42  |  71/38  |  69/35  |  65/30  |
> 	|  pour water  |  38/20  |  86/64  |  84/61  |  82/59  |  80/55  |  75/43  |  73/37  |  72/27  |
>
> 	We agree that, in real-world settings, errors in the upstream 3D perception pipeline can affect the quality of the constructed safety costs and gradients. However, this perception stack is an upstream tool for grounding constraints. Our paper focuses on the orthogonal problem of how to use such safety costs to align a pretrained diffusion policy through post-training, so that the final policy no longer requires privileged information at inference time. Moreover, as perception modules improve or additional sensors are introduced, the same alignment framework can directly benefit without changing the core method.
>
> - **Q1 Concerns on safety violation requiring macroscopic correction**
>
> 	Thank you for the valuable question on constraint injection. As shown in Fig. 3, we would like to humbly clarify that our method is a post-training alignment approach that seeks a  **safe mode within the neighbourhood of the base policy distribution**, rather than performing large trajectory replanning. Accordingly, the final-few-step injection is intended for  **safe mode selection and refinement**. If avoiding a hazard requires a fundamentally different macroscopic strategy absent from the pretrained policy, then our method is not intended to synthesize that behavior from scratch, since no external utility signals are available in the alignment setting. We will clarify this more explicitly in the revision.
>
> - **Q3 Risk of Over-Conservative Solutions**
>
> 	We thank the reviewer for this important question. In principle, a conservative local optimum is possible if safety and task success are fundamentally misaligned. However, our method mitigates this risk through the KL minimization in Eq. (5). It keeps the aligned policy close to the pretrained policy and therefore acts as a trust-region constraint around an already task-competent behavior distribution. This discourages collapse to a trivially safe but ineffective policy; formally, the task performance is bounded as
>
>     $$\left| J_r(\pi) - J_r(\mu) \right| \le \frac{2R_{\max}}{(1-\gamma)^2} \sqrt{\frac{1}{2}\sup_s D_{\mathrm{KL}}\big(\pi(\cdot|s)\|\mu(\cdot|s)\big)},$$
>
>     where $J_r(\pi)$ is the discounted task return under reward $r$, $R_{\max}$ bounds the per-step reward, and $\gamma$ is the discount factor [1,2].
>
>     As for the possibility that the safe region is disconnected from the task-success region, this scenario appears unlikely in our settings. Such a disconnect would suggest that the pretrained policy, and  **its underlying demonstrations for training, are largely unsafe**, which is inconsistent with the assumption of a competent pretrained manipulation policy. We will include these analyses into the revised paper.
>
> **References:**
>
> [1] Kakade, Sham, and John Langford. "Approximately optimal approximate reinforcement learning."  _Proceedings of the nineteenth international conference on machine learning_. 2002.
>
> [2] Schulman, John, et al. "Trust region policy optimization."  _International conference on machine learning_. PMLR, 2015.

---

> > ### Author Rebuttal · Reviewer_sz4p · 2026-04-04
> >
> > Thank you for your response. I will keep my current recommendation.

---

> > > ### Author Response · Authors · 2026-04-05
> > >
> > > We are very glad for learning that your concerns have been resolved. We remain fully available for any new questions arise during the discussion phase. Moreover, we would be incredibly grateful if you might consider adjusting your score upwards to reflect this positive resolution. Thanks for your consideration again！

---

### Official Review · Reviewer_Dghk · 2026-03-13

**Soundness:** 3
**Presentation:** 4
**Significance:** 3
**Originality:** 3
**Overall Recommendation:** 5
**Confidence:** 4

**Summary:**

This paper introduced a self-evolving post-training method that aligns pre-trained diffusion-based imitation learning policy with additional physical constraints such as unexpected contacts and object falling. The major novelties include the constrainted diffusion policy alignment and curriculum-based policy distillation. Compared to the mainstream RL-based finetuneing, the authors claim that the proposed method show advanced sample efficiency and more stable alignment process. The method has been evaluated on simulation-based benchmarks and a few highly challenging real-world tasks with rich ablations and task details.

**Compliance With Llm Reviewing Policy:**

Affirmed.

**Final Justification:**

The rebuttal has addressed most of my concerns.

**Key Questions For Authors:**

1. On implementation, what is the major difference between a naive reward function based on the states and the approximated score for Curriculum Implicit Safety Teacher computed with Eq. (15)? it still seems some form of "reward function" to me.
2. Can you explain why the water cap is on but in line 1553 it says "which requires the bottle to be held directly above the cup to avoid spillage during the pouring phase." What is the real experimental setting on this task?

**Limitations:**

No limitations are mentioned. Apparently, the safety criteria are still relied on human experts, which is very similar to reward engineering. Additional discussion on computational cost, rollout cost and so on will be appreciated.

**Strengths And Weaknesses:**

Strenght:
1. The idea of framing the post-training alignment as a constrained policy distillation is very interesting to me. This provides a novel direction for injecting additional constraints (safety constraints, etc.) or additional preference to a pre-trained policy without reward engineering and RL finetuning.
2. The technical claims and methodology looks reasonable and understandable. Base on the results, the proposed method seems very effective and sample efficient
3. The real world experiments are challenging, especially the heat sink installation. The success rate after distillation seems very good.
4. The paper provide sufficient ablation studies in both main text and appendix on each design item of their method, which is clear and effective.
5. The overall delivery of the paper is good with beautiful illustrations, plots, and figures. The narrative is clear and easy to follow.

Weakness:
1. The definition of success and safety remains unclear to me, in my opinion in many cases they are correlated and Figure 8 shows such correlation.
2. The water bottle design seems flawed. Why does it fail to pour correctly when the cap is still on?
3. Comparison to RL finetuning on Diffusion-based pretrained policy seems missing? There are plenty of such works (e.g., DPPO, FDPP, NCDPO, etc.)
4. Further analysis on the difference between policy distillation and RL finetuning will be appreciated

---

> ### Author Rebuttal · Authors · 2026-03-31
>
> - **W1 Distinction in definition between success and safety**
>
>     Thanks for this thoughtful comment. We agree that success and safety are correlated, but they are conceptually distinct: success measures task completion, while safety avoids predefined unsafe interactions (Appx. D.3). For instance, a rollout may complete a task with unsafe contact (e.g., poking an egg before successfully grasping it), or remain safe but fail (e.g., not reaching any object). We will clarify this distinction in the revision.
>
> - **W2 & Q2 unscrewed Water Cap**
>
>     We thank the reviewer for pointing it out and apologize for the unclear illustration. In the real-world experiment, we do not perform actual liquid pouring. Instead, we use a capped bottle as a proxy task, where the robot must place the bottle above the cup and maintain a pouring pose with a tilt angle above 60° (Appx. D.3). This reduces **reset overhead and improves consistency**, since real pouring would introduce variability from refilling, spillage, and cleaning. The constraint is defined by bottle alignment above the cup during the pouring phase as a proxy for spill-related misalignment. We will revise the text and caption accordingly.
>
> - **W3 Comparison to RL-based baselines**
>
>     Thanks for this valuable suggestion. We also compare against the original DPPO and FDPP. The results support the same conclusion: PACT performs better in the safety-alignment setting, which we attribute to directly distilling safety gradients rather than relying on sparse rewards. We will include these results in the revision.
>
> 	| Method  | pick dual bottles | handover apple | stack blocks | pour water |
> 	|:-:|:-:|:-:|:-:|:-:|
> 	| Base | 67/36 | 63/64 | 63/24 | 38/20 |
> 	| DPPO  | 69/35 | 68/67 | 71/32  | 58/26 |
> 	| FDPP  | 70/38 | 65/66 | 72/34 | 57/29 |
> 	| PACT  | **96/89** | **81/82** | **80/48** | **86/64** |
>
> 	We would like to humbly clarify that original experiments already covered several diffusion RL baselines, including improved PPO, AWR, QSM, and DIPO, with details in Appx. C.7. Our PPO implementation follows a stronger DPPO-based recipe with improved ratio estimation and extra stabilization [1].
>
> - **W4 Further Discussion on policy distillation and RL**
> **1) supervision signal.** RL-based methods typically treat safety cost as a reward and estimate policy gradients from samples batch, making optimization closer to a zero-order procedure. As a result, updates are often noisy and sensitive to sample quality. In contrast, PACT uses the gradient of the safety cost directly, yielding a more direct and stable first-order update toward the feasible region.
> **2) solver flexibility during sampling.** Many diffusion RL methods are tied to specific first-order samplers due to likelihood-ratio estimation or related assumptions,whereas our distillation objective is solver-agnostic and supports efficient rollout with few denoising steps and higher-order solvers.
>
> - **Q1 Difference between reward function and score**
>
>     Thanks for the question. We respectfully clarify that Eq. (15) differs from naive reward in RL. A reward is a **scalar signal** that requires RL machinery such as likelihood estimation to update the policy. In contrast, Eq. (15) defines an approximate score correction in the diffusion process. It converts the safety cost into a **vector-valued first-order gradient** on the noisy action, adjusting the denoising direction toward safer regions. This yields dense supervision at every denoising step and can be distilled by simple regression.
>
> - **L1 Manual-designed Safe Criteria**
>
>     Thanks for pointing this out. We agree that specifying safety criteria requires domain knowledge, but this differs from reward engineering, which designs scalar objectives to induce **task-specific** desired behavior. In contrast, safety costs (e.g., poking) are **shared among tasks** and can be automatically synthesized with VLMs [2] and unsafety taxonomies [3] in a scalable way. Our contribution is orthogonal: we study how to align a pretrained diffusion policy to given safety constraints via post-training rather than proposing a new constraint design pipeline.
>
> - **L2 Discussion on Computation & Rollout Costs**
>
>     PACT does not rely on human demonstrations, task rewards, or test-time guardrails. It improves the policy using self-collected rollouts to enforce constraints. Concretely, PACT uses 8 training iterations with 40 rollouts each, corresponding to \~45 GPU hours (~24G VRAM occupied) for every real-world task. Additional details on cost are provided in Appx. C.5 and D.4.
>
> **References:**
>
> [1] Intelligence, Physical, et al. "$\pi^{*}_{0.6}$: a VLA That Learns From Experience."  _arXiv_ _preprint arXiv:2511.14759_(2025).
>
> [2] Huang, Wenlong, et al. "ReKep: Spatio-Temporal Reasoning of Relational Keypoint Constraints for Robotic Manipulation."  CoRL 2024.
>
> [3] Deng, Haoyuan, et al. "SafeBimanual: Diffusion-based trajectory optimization for safe bimanual manipulation."  CoRL 2025.

---

> > ### Author Rebuttal · Reviewer_Dghk · 2026-04-04
> >
> > The rebuttal has addressed most of my concerns. It's a very interesting paper to be accepted.

---

> > > ### Author Response · Authors · 2026-04-05
> > >
> > > Thank you for the thoughtful follow up and improving your score. We are very grateful that our rebuttal addressed your concerns, and we sincerely appreciate your positive assessment of the paper. We remain happy to answer any further questions during the discussion period.

---

### Official Review · Reviewer_R7Sy · 2026-03-13

**Soundness:** 3
**Presentation:** 3
**Significance:** 3
**Originality:** 3
**Overall Recommendation:** 5
**Confidence:** 4

**Summary:**

This paper proposes a projection-based safety method for diffusion-based robot motion planning. The idea is to supervise the diffusion process at test time with gradients derived from safety constraints; to mitigate the safety/performance tradeoff, the proposed method uses a tightening safety curriculum, which is proven to monotonically improve the safety cost associated with each task. The method increases success and safety rates in comparison to a variety of baselines.

**Compliance With Llm Reviewing Policy:**

Affirmed.

**Final Justification:**

I think the paper is good, and the rebuttal addressed my major concerns.

**Key Questions For Authors:**

* Is the safe rate on hardware evaluated manually?

* Maybe I missed it, but what is the average inference time of the method? How much does the proposed approach impact computation time?

**Limitations:**

The paper has no discussion of limitations that I could find; the conclusion is very sparse. The authors sufficiently address potential negative social impact, but I would like to see an explicit subsection explaining the method's weaknesses in the authors' view.

**Strengths And Weaknesses:**

Strengths:
* The paper addresses an important problem of injecting safety into cutting-edge IL models
* The paper is well written and pretty easy to read, and the figures are nice, especially Fig. 3
* The "self-evolving" system of tightening constraint enforcement is a good idea to handle the safety/performance tradeoff; the paper shows that this tradeoff is not in fact as hard as people might think
* Incorporating safety (or other gradient sources) into diffusion while running in real time is hard [R1], so the efficiency claimed in this paper seems very good
* The paper includes real-world examples
* The paper spans theory and implementation, which is always nice to see

Weaknesses:
* The main weakness is I don't see a route to enforcing hard constraints with this method
* The method has not been compared, to my knowledge, against very relevant recent baselines [R1,R2] (at least a literature comparison should suffice)
* The implementation of physical constraints via cost functions seems a bit heuristic and situation-specific

References:

[R1] Jung, W., Mishra, U.A., Arachchige, N.R., Chen, Y., Xu, D. and Kousik, S., Joint Model-based Model-free Diffusion for Planning with Constraints. In 9th Annual Conference on Robot Learning.

[R2] Nakamura, K., Peters, L. and Bajcsy, A., 2025. Generalizing safety beyond collision-avoidance via latent-space reachability analysis. In Robotics: Science and Systems XXI.

---

> ### Author Rebuttal · Authors · 2026-03-31
>
> - **W1 Hard constraints enforcement**
>
>     Thanks for this question. Our method can be extended to hard constraints by replacing the soft constraints of Eq. (4) with hard ones, which removes the expectation form and yields a more restrictive feasible policy set $\Pi_h=\\{\pi: c_k(s,a)\le d_k,\ \forall k, a \sim \pi(\cdot | s)\\}.$ We can solve the problem with Feasibility-Dependent Optimization [1], which decomposes the problems into feasible and infeasible regions:
>
>     $$\text{Feasible:}\min_{\pi} \mathbb{E}\_s\left[D_{\mathrm{KL}}(\pi | \mu_{\phi})\cdot \mathbb{I}_{s\in \mathcal{S}_h}\right] \quad \text{s.t.}\ \pi \in \Pi_h(s),\ \forall s \in \mathcal{S}_h$$
>
>     $$\text{Infeasible:} \min_{\pi}\ \mathbb{E}\_s \left[c_k(s, a) \cdot \mathbb{I}_{s\notin \mathcal{S}_h}\right]$$
>
>     Where $\mathcal{S}\_{h}$ is the feasible region including all feasible states $\mathcal{S}\_{h}=\\{s | c_{k}(s, a) \le d_k, \forall k, a \sim \pi(\cdot | s)\\}$. Within this framework, our distillation strategy can still be used to reduce violations while preserving pretrained behavior in safe regions. Prior work [1] also provides a theoretical characterization of the safe optimum in this hard constraints setting. We will add this discussion in the revision.
>
> - **W2 Compare against baselines**
>
>     Thanks for valuable suggestion on more rigorous evaluations. Firstly, Jung et al. propose JM2D, an inference-time guardrail method to achieve mutual compatibility [R1]. As no official implementation is available, we made a careful best-effort reproduction based on the paper. The results (succ. / safe) are shown below.
>
>     | Method | pick dual bottles | handover apple | stack blocks | pour water |
>     |-|-|-|-|-|
>     | Base  | 67/36 | 63/64 | 63/24 | 38/20 |
>     | JM2D  | 87/66 | 65/68 | 76/27 | 75/40 |
>     | PACT | **96/89** | **81/82** | **80/48** | **86/64** |
>
>     Second, Nakamura et al. propose computing safety costs in the latent space of a generative world model [R2]. This direction is complementary to ours and could be integrated into PACT as an alternative cost representation. We will expand the baseline and related-work discussion in the revision.
>
> - **W3 Physical constraints via cost functions seems situation-specific**
>
>     Thanks for pointing it out. We agree that the concrete form of a physical constraint is often task-dependent, but this is inherent to robotic safety rather than specific to our method. In robotics, translating safety requirements into computable cost functions is a general practice, and prior work has explored scalable ways to derive such costs with VLMs [2] or unsafety taxonomies [3]. Following this paradigm, we adopt an existing unsafe interaction taxonomy and instantiate the corresponding costs automatically (Appx. B.2).
>         Meanwhile, we aim to humbly clarify that our contribution is orthogonal: rather than proposing a new constraint-engineering pipeline, we study how to **align a pretrained diffusion policy to satisfy given safety constraints through post-training**. The fact that our method works across multiple tasks and heterogeneous cost types further suggests its compatibility.
>
> - **Q1 Manual evaluation for safe rate**
>
>     In real-world experiments, safe rate is manually evaluated using predefined criteria (Fig. 8, Appx. D.3) to ensure an objective protocol. Annotators judge safety and success accordingly following common practice as concurrent works. We will clarify this process in the revision.
>
> - **Q2 Average inference time & Impact of proposed approach**
> We thank the reviewer for this question. A key advantage of PACT is that safety alignment is done entirely during post-training, not deployment. Thus, the aligned policy uses the same inference procedure as the base policy without extra test-time overhead, so the inference time remains unchanged. Inference times per action chunk for all policies are listed below:
> 	| Model | Inference Time (ms) |
> 	|-|-|
> 	| DP | 455.53 |
> 	| DP3 | 49.84 |
> 	| RDT | 179.43 |
> 	| $\pi_{0.5}$ | 91.83 |
>
>     We will update the results with analyses in the revised version.
>
> - **Limitations**
>
>     While PACT effectively enforces safety constraints in diffusion-based manipulation, it possesses several limitations: its safety alignment relies on the accuracy of upstream vision models, which can be sensitive to environmental occlusions and the post-training phase involves extra computational overhead due to the need for self-rollouts. We will integrate a more comprehensive discussion of the limitations and enriched conclusion in the revised paper.
>
> **References:**
>
> [1] Zheng, Yinan, et al. "Safe Offline Reinforcement Learning with Feasibility-Guided Diffusion Model."  ICLR 2024.
>
> [2] Huang, Wenlong, et al. "ReKep: Spatio-Temporal Reasoning of Relational Keypoint Constraints for Robotic Manipulation."  CoRL 2024.
>
> [3] Deng, Haoyuan, et al. "SafeBimanual: Diffusion-based trajectory optimization for safe bimanual manipulation."  CoRL 2025.

---

> > ### Author Rebuttal · Reviewer_R7Sy · 2026-03-31
> >
> > I really appreciate the authors' efforts on this rebuttal. They have answered my concerns fully, and I think this paper is very interesting.

---

> > > ### Author Response · Authors · 2026-04-01
> > >
> > > Thank you for your positive feedback and for confirming that our rebuttal addressed your concerns. We are grateful for your consideration in updating the score to reflect your current evaluation. We also remain happy to address any further questions or suggestions during the discussion phase.

---

### Decision · Program_Chairs · 2026-04-30

**Decision:**

Accept (spotlight)

**Comment:**

This paper proposes PACT, a post-training framework for aligning pretrained diffusion-based manipulation policies with physical safety constraints. The method distills constraint gradients into the diffusion model via a reverse-KL objective and employs a progressive curriculum to prevent catastrophic forgetting, with formal guarantees of monotonic safety improvement and bounded policy shift.

All four reviewers see clear merit in this work and recommend acceptance. Three reviewers gave a score of 5 (Accept) and one gave a 4 (Weak Accept). The consensus is that the paper addresses an important and timely problem, and are all in favor of acceptance.

The main concerns raised during review — including missing comparisons to recent baselines (JM2D, DPPO, FDPP), robustness to perception noise, scalability in low-data regimes, and the distinction between success and safety metrics — were all addressed satisfactorily in the rebuttal. The authors provided additional experiments and clarifications that resolved the concerns of all reviewers.

I recommend a strong accept.